# Jacobian-based Causal Discovery with Nonlinear ICA

**Patrik Reizinger**\*                                            *patrik.reizinger@uni-tuebingen.de*
*University of Tübingen, Germany*
*International Max Planck Research School for Intelligent Systems (IMPRS-IS)*
*European Laboratory for Learning and Intelligent Systems (ELLIS)*

**Yash Sharma**                                                  *yash.sharma@uni-tuebingen.de*
*University of Tübingen, Germany*
*International Max Planck Research School for Intelligent Systems (IMPRS-IS)*

**Matthias Bethge**                                            *matthias.bethge@uni-tuebingen.de*
*University of Tübingen, Germany*

**Bernhard Schölkopf**                                                  *bs@tuebingen.mpg.de*
*Max Planck Institute for Intelligent Systems, Tübingen, Germany*

**Ferenc Huszár**†                                                        *fh277@cam.ac.uk*
*University of Cambridge, United Kingdom*

**Wieland Brendel**†                                            *wieland.brendel@tuebingen.mpg.de*
*Max Planck Institute for Intelligent Systems, Tübingen, Germany*

**Reviewed on OpenReview:** *https://openreview.net/forum?id=2Yo9xqR6Ab*

## Abstract

Today's methods for uncovering causal relationships from observational data either constrain functional assignments (linearity/additive noise assumptions) or the data generating process (e.g., non–i.i.d. assumptions). Unlike previous works, which use conditional independence tests, we rely on the inference function's Jacobian to determine nonlinear cause-effect relationships. We prove that, under strong identifiability, the inference function's Jacobian captures the sparsity structure of the causal graph; thus, generalizing the classic LiNGAM method to the nonlinear case. We use nonlinear Independent Component Analysis (ICA) to infer the underlying sources from the observed variables and show how nonlinear ICA is compatible with causal discovery via non–i.i.d. data. Our approach avoids the cost of exponentially many independence tests and makes our method end-to-end differentiable. We demonstrate that the proposed method can infer the causal graph on multiple synthetic data sets, and in most scenarios outperforms previous work.

## 1 Introduction

Traditional statistical learning methods model correlations in data. Though they have achieved super-human performance in multiple fields, they have limited value in understanding cause-effect relationships. A prevalent consequence of this shortcoming is the models' tendency to learn from spurious features or shortcuts (Geirhos et al., 2020) (e.g., classifying objects based on their backgrounds). In contrast, *causal models* construct the world according to the Independent Causal Mechanisms (ICM) principle (Peters et al., 2017), where building blocks (mechanisms) neither influence nor inform each other. Modeling temperature $T$ and altitude $A$ is a classic example (Peters et al., 2017): changing $A$ affects $T$, but not vice versa—this relationship is described by the Directed Acyclic Graph (DAG) $A \rightarrow T$. The ICM principle means that the same mechanism $p(T|A)$ describes how altitude affects temperature for different $p(A)$, but the same cannot be said about $p(A|T)$ and $p(T)$. *Causal Discovery (CD)* describes the process of extracting causal structure from data in the form of a DAG. Having *interventional* data—such as in the form of Randomized Controlled Trials (RCTs)—is desirable as it enables answering questions of interventional nature, such as 'What will happen if variable $X$ is changed?'

---

\*Corresponding author. Code available at: `github.com/rpatrik96/nl-causal-representations`
†Joint supervision

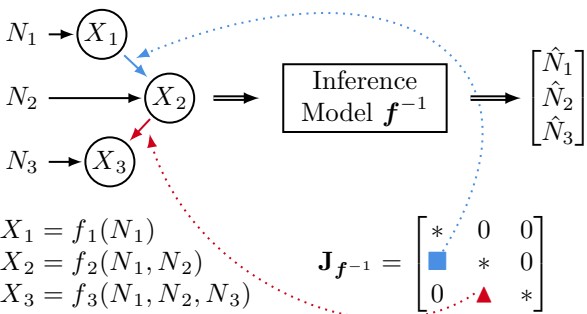

Figure 1: **The Jacobian of the inference network $\mathbf{J}_{f^{-1}}$ informs about the DAG**. We show that if observations $\boldsymbol{X}$ are generated from noise variables $\boldsymbol{N}$ via a general nonlinear Structural Equation Model (SEM) $\boldsymbol{f}$, then the corresponding DAG can be inferred from the Jacobian of a model that identifies $\boldsymbol{N}$ under certain assumptions on $\boldsymbol{N}$

However, RCTs can be costly, infeasible (Eberhardt et al., 2005), or even unethical. Thus, developing effective CD methods reliant on *observational* data alone is of significant interest. In general, inferring the causal direction is provably impossible without additional constraints or assumptions (Zhang et al., 2015); therefore, existing methods constrain either the model class (i.e., the functions generating the observations) or the data distribution. On the model side, these constraints include linear (Shimizu et al., 2006; Tashiro et al., 2014; Shahbazinia et al., 2021; Zheng et al., 2018) or specific nonlinear relationships (e.g., with additive noise) (Hoyer et al., 2008; Peters et al., 2011; Schölkopf et al., 2021a; Yu et al., 2019; Shen et al., 2022; Lachapelle et al., 2020; Ng et al., 2022). On the data side, assumptions include non-stationarity (Hyvärinen & Morioka, 2016; Monti et al., 2019) or exchangeability (Guo et al., 2022).

CD aims to infer the ground-truth cause-effect relationships, which connects it to the *identifiability* literature, where the goal is to learn a model equivalent to the ground truth (up to indeterminacies, such as permutations or element-wise nonlinearities).

An extensively studied method for learning identifiable representations is Independent Component Analysis (ICA) (Comon, 1994; Hyvärinen et al., 2001), which requires that the inferred components *(sources)* are independent. Recent work has relied on nonlinear Independent Component Analysis (NLICA) for identifiability (Zimmermann et al., 2021; Klindt et al., 2021; Hyvärinen & Morioka, 2016; Willetts & Paige, 2021; Khemakhem et al., 2020a; Hyvärinen et al., 2019; Morioka et al., 2021; Monti et al., 2019; Khemakhem et al., 2020b; Gresele et al., 2019; Hyvärinen & Morioka, 2017; Hyvärinen et al., 2010; Hälvä & Hyvärinen, 2020; Lachapelle et al., 2022).

Instead of using pairwise independence tests, we draw inspiration from the Linear Non-Gaussian Acyclic Model (LiNGAM) (Shimizu et al., 2006), which uses a weight matrix to infer the DAG of a linear causal model. We extend this approach to the nonlinear case by showing that the Jacobian of the *ground-truth* inverse Data Generating Process (DGP) (mapping from observations $\boldsymbol{X}$ to noise variables $\boldsymbol{N}$) captures the sparsity structure of the DAG (Prop. 1). Since the ground truth model is generally unknown, we transfer our insight to the Jacobian of the learned *inference model*[1] (i.e., the empirical estimate of the ground-truth $\boldsymbol{X} \to \boldsymbol{N}$ map; cf. Prop. 2). There, we quantify the requirements on the inference model with the notion of strong identifiability is fulfilled (Khemakhem et al., 2020b, Def. 2) (cf. Defn. B.1) and show that causal models provide an inductive bias to resolve the permutation indeterminacy (Lem. 1). We guarantee identifiability via NLICA; thus, our work is akin to the NonSENS method (Monti et al., 2019), which showed that NLICA can be used for bivariate CD with general nonlinear functions and non–i.i.d. observational data. However, our proposal works in the multivariable case. Relying on the Jacobian removes the cost of $d^2$ independence tests for a DAG with $d$ nodes. However, with current NLICA methods, we could only scale up to ten nodes.

Our **contributions** can be summarized as follows:

---

[1]In our paper, inference refers only to this process and not to amortized inference for direct graph discovery as proposed in Lorch et al. (2022)

1. We prove that the inverse DGP's Jacobian encodes the DAG structure (Prop. 1);
2. We show that causal models allow us to resolve the permutation indeterminacy of ICA (Lem. 1);
3. Our **main result** (Prop. 2) proves that we can infer the DAG from the Jacobian of the inference function, while removing the need for independence tests;
4. We propose an end-to-end multivariable CD method for nonlinear functions from observational but non–i.i.d. data and show how contrastive NLICA is compatible with CD;
5. We experimentally show that our proposed method can infer the DAG across multiple synthetic data sets.

## 2 Background

Here, we describe causal models and connect their estimation to ICA and defer the details to Appx. A.

**Structural Equation Models (SEMs).** Given $d$-dimensional observed $\boldsymbol{X}=(X_1,\ldots,X_d)$ and noise (independent) variables $\boldsymbol{N}=(N_1,\ldots,N_d)$, their causal relationship is given by $d$ *deterministic* functional assignments (Pearl, 2009), constituting the generative model:

$$X_i := f_i\left(\boldsymbol{Pa}_i, N_i\right) \qquad \forall i, \tag{1}$$

where $\boldsymbol{Pa}_i \subset \boldsymbol{X}$ are the parents of $X_i$ and $f_i$ are the components of the vector-valued function $\boldsymbol{f}$. We describe the computation of $\boldsymbol{X}$ for a given $\boldsymbol{N}$ with an iterative process (denoting the iteration step with a superscript), which is a useful concept for justifying our proposal (§ 3). Initially, $\boldsymbol{N}$ is drawn from its density. To calculate $\boldsymbol{X}$ for $\boldsymbol{N}$, the functional assignment $\boldsymbol{f}$ needs to be applied $d$ times. Namely, according to (1), each $X_i$ requires that its parents $\boldsymbol{Pa}_i$ are calculated. After sampling $\boldsymbol{N}$, only the (empty) parent sets of root nodes are calculated. Thus, the first application of $\boldsymbol{f}$ yields the $X_i$ values for such nodes. In the second iteration, the children of root nodes can be calculated (since we have all parents from the first iteration), and so on. This yields an iterative algorithmic formulation of the SEM, describing the computational graph given by the DAG as:

$$\boldsymbol{X} = \boldsymbol{X}^d := \boldsymbol{f}^{(d)}\left(\boldsymbol{X}^0, \boldsymbol{N}\right) = \boldsymbol{f}\left(\boldsymbol{X}^{(d-1)}, \boldsymbol{N}\right) = \boldsymbol{f}\left(\boldsymbol{f}\ldots\left(\boldsymbol{f}\left(\boldsymbol{X}^0, \boldsymbol{N}\right), \boldsymbol{N}\right), \boldsymbol{N}\right), \tag{2}$$

where $\boldsymbol{X}^0$ is the initial value (w.l.o.g., we assume $\boldsymbol{X}^0 = \boldsymbol{0}$, since calculating the functional assignments will overwrite every $X_i$). We will also denote $\boldsymbol{X} = \boldsymbol{X}(\boldsymbol{N})$ to indicate that $\boldsymbol{X}$ is *deterministically* determined by a particular $\boldsymbol{N}$. As in most previous works (Vowels et al., 2022, Table 1), we assume *no confounders* (all variables are observed) and *faithfulness* (loosely speaking, the coefficients/functions will not cancel an edge, cf. Assum. 1).

**Causal Discovery (CD).** In CD, the data is assumed to be generated by a causal process, and the aim is to infer the corresponding DAG, which enables reasoning about interventions (without the DAG, the joint distribution $p(\boldsymbol{N})$ only admits observational queries) (Peters et al., 2017; Pearl, 2009). Algorithmic approaches include combinatoric search (Shimizu et al., 2006; Hoyer et al., 2008; Hyttinen et al., 2013; Mitrovic et al., 2018; Raskutti & Uhler, 2018; Spirtes et al., 2000; Vowels et al., 2022), continuous optimization (Zheng et al., 2018; Lee et al., 2019; Wei et al., 2020; Ng et al., 2020; Vowels et al., 2022), and neural networks (Yu et al., 2019; Ng et al., 2022; Khemakhem et al., 2021; Yang et al., 2021; Goudet et al., 2018; Kalainathan et al., 2018; Vowels et al., 2022; Kyono et al., 2020; Moraffah et al., 2020)—we focus on the latter. Zhang et al. (2015) proved that identifying the causal direction in a general SEM is impossible without constraints on the function class and/or data distribution.
Functional constraints can include linear (Shimizu et al., 2006; Zheng et al., 2018; Squires et al., 2023), additive nonlinear ($X_i = f_i(\boldsymbol{Pa}_i) + N_i$) (Hoyer et al., 2008; Ng et al., 2022; Lachapelle et al., 2020; Schölkopf et al., 2021a; Yang et al., 2021), affine nonlinear ($X_i = f_i(\boldsymbol{Pa}_i) + h_i(N_i)$) (Khemakhem et al., 2021; Shen et al., 2022), or polynomial (Ahuja et al., 2022b) models. Regarding the data distribution, some models require access to interventions (Brouillard et al., 2020; Ke et al., 2020; Lippe et al., 2021; Ahuja et al., 2022b); others assume that $\boldsymbol{N}$ is Gaussian (Kalainathan et al., 2018; Lachapelle et al., 2020) or non-Gaussian (Shimizu et al., 2006); or require non-stationarity (Monti et al., 2019), exchangeability (Guo et al., 2022), or discreteness (Ke et al., 2020) of $\boldsymbol{N}$. Variational-inference–based formulations require a prior over the DAGs (Lorch et al., 2021; 2022; Charpentier et al., 2022) or utilize labels (Yang et al., 2021). Our work was inspired by (Monti et al., 2019), which provides a bivariate CD method for general nonlinear functions and non-stationary data. The authors leverage recent results in NLICA (cf. next section for details) to identify the causal direction.

Although they demonstrate applicability to multivariable problems, the use of pairwise independence tests constrains scalability. In our work, we extend these results with an end-to-end solution in § 3.

**Identifiability and ICA.** Independent Component Analysis (ICA) (Comon, 1994; Hyvärinen et al., 2001) models the observed variables $\boldsymbol{X}$ as a mixture of *independent* variables $\boldsymbol{N}$ via a deterministic function $\boldsymbol{f}$, and focuses on defining models that are *identifiable*—i.e., $\boldsymbol{N}$ can be recovered up to indeterminacies (e.g., scaling, permutation, sign flips, element-wise transformations). Since this is provably impossible in the nonlinear case without further assumptions (Darmois, 1951; Hyvärinen & Pajunen, 1999; Locatello et al., 2019), recent work has focused on incorporating *auxiliary* variables (Hyvärinen et al., 2019; Gresele et al., 2019; Khemakhem et al., 2020a; Gassiat et al., 2022), exploiting temporal structure in the data (Hyvärinen & Morioka, 2017; 2016; Hälvä & Hyvärinen, 2020; Morioka et al., 2021; Monti et al., 2019; Hyvärinen et al., 2010; Klindt et al., 2021; Zimmermann et al., 2021), or restricting the model class (Shimizu et al., 2006; Hoyer et al., 2008; Zhang & Hyvärinen, 2009; Gresele et al., 2021). Several works have related (nonlinear) ICA to SEM estimation (Gresele et al., 2021; Monti et al., 2019; Shimizu et al., 2006; Von Kügelgen et al., 2021; Hyvärinen et al., 2023) by inverting the DGP—i.e., estimating $\boldsymbol{f}^{-1}$ with an *inference model* $\widehat{\boldsymbol{f}}^{-1}$.

## 3 Inferring causal structure from Jacobians

### 3.1 Intuition

The method we propose can be intuitively understood as a nonlinear extension of LiNGAM (Shimizu et al., 2006; Hoyer et al., 2008; Peters et al., 2011). LiNGAM assumes a linear causal relationship between observations $\boldsymbol{X}$ and the noise variables $\boldsymbol{N}$, i.e., $\boldsymbol{X} = \mathbf{W}\boldsymbol{N}$. Since the noise variables are assumed to be statistically independent, linear ICA can uncover the (non-Gaussian) sources $\boldsymbol{N}$ from the observations $\boldsymbol{X}$, which allows us to extract the DAG from $\mathbf{W}^{-1}$ as we show in the following example.

**Example 1** (Motivating example for linear SEMs)**.** *Assume a linear causal model with three variables, the DAG $X_1 \to X_2 \to X_3$, and functional relationships: $X_1 := N_1$; $X_2 := aX_1 + N_2$; $X_3 := bX_2 + N_3 : a, b \in \mathbb{R} \setminus \{0\}$. The DGP generates samples according to the DAG and has the matrix form on the left—we focus on the elements below the main diagonal as for recovering the DAG, only the paths (i.e., series of directed edges) between $X_i$ and $X_j$ are required and the main diagonal expresses the $N_i \to X_i$ edges. Inverting the DGP with an inference model (i.e., expressing $N_i$ as a function of $X_j$; LiNGAM uses ICA to estimate the DGP) yields the matrix on the right with elements below the main diagonal capturing the DAG's $X_i \to X_j$ edges (as shown by color coding):*

$$\begin{bmatrix} X_1 \\ X_2 \\ X_3 \end{bmatrix} = \begin{bmatrix} 1 & 0 & 0 \\ a & 1 & 0 \\ ab & b & 1 \end{bmatrix} \begin{bmatrix} N_1 \\ N_2 \\ N_3 \end{bmatrix}; \qquad \begin{bmatrix} N_1 \\ N_2 \\ N_3 \end{bmatrix} = \begin{bmatrix} 1 & 0 & 0 \\ -a & 1 & 0 \\ 0 & -b & 1 \end{bmatrix} \begin{bmatrix} X_1 \\ X_2 \\ X_3 \end{bmatrix}$$

**Overview of theoretical results.** Our method extends LiNGAM to nonlinear DGPs. First, we show that the inverse DGP's Jacobian and the DAG structure are structurally equivalent (Prop. 1). To apply Prop. 1 to a learned inference model, we describe up to what indeterminacies the inference model is need to be known. Because the ground-truth DGP can only be identified up to certain indeterminacies like scaling, permutation, and sign flips, we need to show for which identifiability notion structural equivalence is preserved (Prop. 2). This requires that we can resolve permutation indeterminacies, which we prove for SEMs in Lem. 1, then design an algorithm for this purpose (§ 3.4).

### 3.2 DAG equivalence

To justify using the Jacobian of $\boldsymbol{f}^{-1}$, i.e., the inverse of the DGP, (denoted as $\mathbf{J}_{\boldsymbol{f}^{-1}}$), akin to LiNGAM's use of a weight matrix, we first connect the DAG and $\mathbf{J}_{\boldsymbol{f}^{-1}}$ via fundamental concepts from graph theory. The *adjacency matrix* $\boldsymbol{\mathcal{A}}$ of a graph with $d$ nodes is a binary $d \times d$ matrix where each matrix element indicates the presence, or absence, of an edge (i.e., a direct connection) between a pair of nodes $X_i, X_j$ (Defn. A.8). The *connectivity matrix* $\boldsymbol{\mathcal{C}}$ of a graph with $d$ nodes is a binary $d \times d$ matrix where each matrix element indicates the presence, or absence, of a *directed path* between two nodes $X_i, X_j$ (Defn. A.9). For DAGs, both $\boldsymbol{\mathcal{A}}$ and $\boldsymbol{\mathcal{C}}$ are *strictly lower-triangular*—this is why we considered only the elements below the main diagonal in Ex. 1. Furthermore, the main diagonal of $\mathbf{J}_{\boldsymbol{f}^{-1}}$ has non-zero elements (Ex. 1). We describe the relationship between $\mathbf{J}_{\boldsymbol{f}^{-1}}$ and $(\mathbf{I}_d - \boldsymbol{\mathcal{A}})$ for a DAG via *structural equivalence*, and investigate its symmetries. Ex. 1 intuits why our claim refers to $\boldsymbol{\mathcal{A}}$ and not $\boldsymbol{\mathcal{C}}$: in the matrix mapping from $\boldsymbol{X}$ to $\boldsymbol{N}$ only the edges (captured by $\boldsymbol{\mathcal{A}}$) are

present. Similar to the linear case (and shown more formally below), $\mathbf{J}_{\boldsymbol{f}^{-1}}$ and $(\mathbf{I}_d - \boldsymbol{\mathcal{A}})$ have the same sparsity structure, meaning $\forall i, j \ (\mathbf{J}_{\boldsymbol{f}^{-1}})_{ij} = 0 \Leftrightarrow (\mathbf{I}_d - \boldsymbol{\mathcal{A}})_{ij} = 0$. We denote this structural equivalence as $\mathbf{J}_{\boldsymbol{f}^{-1}} \sim_{DAG} (\mathbf{I}_d - \boldsymbol{\mathcal{A}})$, with the full definition and its properties formalized as:

**Definition 1** ($\sim_{DAG}$). *Two matrices* $\mathbf{S}, \mathbf{R}$ *of same dimensions are structurally equivalent if* $(\mathbf{S})_{ij} = 0 \iff (\mathbf{R})_{ij} = 0 : \forall i, j;$. *Structural equivalence, denoted as* $\sim_{DAG}$, *has the following properties* ($\circ$ *denotes composition*):

   (i) **D-*invariance:*** *a non-singular diagonal matrix* $\mathbf{D}$ *preserves the sparsity structure; thus,* $(\mathbf{D} \circ \mathbf{S}) \sim_{DAG} \mathbf{S}$

   (ii) $h_0$-***invariance:*** *for zero-preserving transformations* $h_0 : (h_0(\mathbf{S}))_{ij} = 0 \iff (\mathbf{S})_{ij} = 0$ *then* $h(\mathbf{S}) \sim_{DAG} \mathbf{S}$

   (iii) $\pi$-***equivariance:*** *a permutation* $\pi$ *affects the positions of zeros; thus, both operands need to be permuted with the same* $\pi$ *to maintain* $\sim_{DAG}$, *i.e.,* $\mathbf{S} \sim_{DAG} \mathbf{R} \iff (\pi \circ \mathbf{S}) \sim_{DAG} (\pi \circ \mathbf{R})$,

   (iv) ***Transitivity:*** $\mathbf{S} \sim_{DAG} \mathbf{P} \land \mathbf{P} \sim_{DAG} \mathbf{R} \implies \mathbf{S} \sim_{DAG} \mathbf{R}$

   (v) ***Commutativity:*** $\mathbf{S} \sim_{DAG} \mathbf{R} \iff \mathbf{R} \sim_{DAG} \mathbf{S}$.

Before proving structural equivalence, we state our assumptions about the SEM:

**Assumption 1** (SEM assumptions). *We assume that the causal DGP fulfils:*

   *(i) The SEM generative model is given by* (1), *for which there exists an underlying DAG;*

   *(ii)* $N_i$ *are jointly independent;*

   *(iii) There are no hidden confounders (faithfulness/stability); moreover, the Jacobians* $\mathbf{J}_{\boldsymbol{f}}$, $\mathbf{J}_{\boldsymbol{f}^{-1}}$ *are structurally faithful (Assum. A.1);*

   *(iv) each* $f_i$ *is bijective; and*

   *(v) each* $X_i$ *depend on* $N_i$ *(i.e.,* $\frac{\partial \boldsymbol{f}(\boldsymbol{X}, \boldsymbol{N})}{\partial \boldsymbol{N}}\big|_{\boldsymbol{X}, \boldsymbol{N}}$ *is diagonal with non-zero elements)*

Relying on the properties of $\sim_{DAG}$, we prove that $\mathbf{J}_{\boldsymbol{f}^{-1}}$ can be used to extract the DAG for nonlinear SEMs under Assum. 1 (akin to the linear case shown in Ex. 1; the proof is deferred to Appx. E.2)

**Proposition 1.** $[\mathbf{J}_{\boldsymbol{f}^{-1}} \sim_{DAG} (\mathbf{I}_d - \boldsymbol{\mathcal{A}})]$*The inverse DGP's Jacobian* $\mathbf{J}_{\boldsymbol{f}^{-1}}$ *is structurally equivalent to* $(\mathbf{I}_d - \boldsymbol{\mathcal{A}})$, *when Assum. 1 holds.*

*Proof (Sketch).* From the iterative formulation of the SEM in eq. (2), we note that $\boldsymbol{X}$ (or more precisely, $\boldsymbol{X}(\boldsymbol{N})$), is a fixpoint of $\boldsymbol{f}$. Thus, when we apply the chain rule to calculate $\mathbf{J}_{\boldsymbol{f}}$, we will only have two types of terms (on both sides), namely:

$$\mathbf{A} := \frac{\partial \boldsymbol{f}(\boldsymbol{X}, \boldsymbol{N})}{\partial \boldsymbol{X}}\Big|_{\boldsymbol{X}, \boldsymbol{N}}; \ \mathbf{B} := \frac{\partial \boldsymbol{f}(\boldsymbol{X}, \boldsymbol{N})}{\partial \boldsymbol{N}}\Big|_{\boldsymbol{X}, \boldsymbol{N}}. \tag{3}$$

This expression leads us to a closed form of $\mathbf{J}_{\boldsymbol{f}}$. Then we apply the inverse function theorem at $(\boldsymbol{X}, \boldsymbol{N})$ to get $\mathbf{J}_{\boldsymbol{f}^{-1}}$. As the last step, we incorporate the indeterminacies—coming from strong identifiability—and show based on the properties of $\sim_{DAG}$ that the statement of the proposition holds. $\qquad \square$

Prop. 1 implies that we can extract the DAG from $\boldsymbol{f}^{-1}$; i.e., we can reason about interventions (cf. § 2). We note that if $\mathbf{B} = \mathbf{I}_d$, then (29) describes Additive Noise Models (ANMs) (Hoyer et al., 2008), whereas when additionally $\mathbf{A}$ is constant, we recover LiNGAM (Shimizu et al., 2006). Prop. 1 assumes that we have access to $\boldsymbol{f}^{-1}$; however, this is a non-trivial assumption. In the following, we investigate to what extent we need to estimate $\boldsymbol{f}^{-1}$ (in form of $\widehat{\boldsymbol{f}}^{-1}$) to exploit Prop. 1—for this, we leverage the notion of identifiability.

## 3.3   Identifiability requirements of $\widehat{\boldsymbol{f}}^{-1}$

The inference model $\widehat{\boldsymbol{f}}^{-1}$ we learn from the observed data generally differs from the true inverse of $\boldsymbol{f}$ up to certain indeterminacies depending on the identifiability guarantees of the (most commonly) NLICA algorithm. This can include scaling, permutation, sign flips, and monotonic element-wise transformations (Hyvärinen et al., 2001; Khemakhem et al., 2020a; Zimmermann et al., 2021). While element-wise transformations such as scaling or sign-flips do not influence the sparsity structure of the Jacobian, permutations break structural equivalence between the Jacobian and the ground-truth adjacency matrix. That is, we need to resolve the permutation indeterminacy to apply Prop. 1 to $\mathbf{J}_{\widehat{\boldsymbol{f}}^{-1}}$. With the right ordering(s)[2], the Jacobian $\mathbf{J}_{\widehat{\boldsymbol{f}}^{-1}}$ features a lower-triangular structure. The following lemma shows that this property determines the ordering of the noise variables such that they yield a lower-triangular Jacobian, i.e., all possible causal orderings that ensure structural equivalence to the ground-truth adjacency matrix (the proof is deferred to Appx. E.1):

---

[2]The causal ordering does not need to be unique, e.g., in the DAG $X_i \leftarrow X_j \rightarrow X_k$ the nodes $X_i$ and $X_k$ are interchangeable

**Lemma 1.** *[DAG DGPs resolve the permutation ambiguity of ICA] When the DGP is a SEM with functional relationships $\boldsymbol{f}$ and an underlying DAG, then the permutation indeterminacy of ICA $\pi_{\mathrm{ICA}}$ can be accounted for such that the Jacobian of the inference network will have a lower-triangular Jacobian, even with unknown causal ordering $\pi$.*

*Proof (Sketch).* Given that the DGP is structured by a DAG, the adjacency matrix $\boldsymbol{\mathcal{A}}$ is lower triangular and Assum. 1 ensures that diagonal elements are nonzero. The permutation indeterminacy of ICA (which is expressed as a left-multiplication, i.e., affects the rows) comprises matrices that do not violate lower-triangularity. This gives us a single permutation (for a unique causal ordering) or a set of permutations, each of which ensures a lower triangular $\boldsymbol{\mathcal{A}}$. $\qquad\square$

We emphasize that Lem. 1 refers to *two permutations*: the permutation indeterminacy of ICA (Lem. 1 makes a claim about this) and the causal ordering of the SEM. These can be thought of as permuting the rows (ICA indeterminacy) and columns (causal ordering) of the inference model's Jacobian. Most importantly, Lem. 1 shows that we can resolve the permutation indeterminacy, leading to the following result:

**Proposition 2** ($\mathbf{J}_{\boldsymbol{f}^{-1}} \sim_{DAG} \mathbf{J}_{\widehat{\boldsymbol{f}}^{-1}}$ for strongly identified $\widehat{\boldsymbol{f}}^{-1}$)**.** *When the inference model $\widehat{\boldsymbol{f}}^{-1}$ is strongly identified in the sense of Defn. B.1, the permutation indeterminacies are resolved, and Assum. 1 holds, then $\mathbf{J}_{\boldsymbol{f}^{-1}} \sim_{DAG} \mathbf{J}_{\widehat{\boldsymbol{f}}^{-1}}$.*

*Proof.* The indeterminacies of strong identifiability (Defn. B.1) include scalings, sign flips, and permutations. By Def. 1(i), $\sim_{DAG}$ is invariant to scalings and sign flips; whereas Def. 1(iii) states equivariance for permutations, but by Lem. 1, those can be resolved for SEMs. $\qquad\square$

By Def. 1(ii), Prop. 2 also holds when indeterminacies include zero-preserving transformations.

### 3.4 Algorithm for CD and determining $\pi$

Based on Lem. 1 and Props. 1 and 2, we propose a two-step approach for extracting the DAG from observational but non–i.i.d. data for general nonlinear $\boldsymbol{f}$ :

1. First, we use a suitable nonlinear ICA algorithm to estimate $\boldsymbol{f}^{-1}$ up to permutations and zero-preserving element-wise nonlinearities with an inference model $\mathbf{J}_{\widehat{\boldsymbol{f}}^{-1}}$.
2. Second, we resolve the permutation indeterminacy by accounting for the causal graph structure.

Regarding the second step, we learn the permutations after training with an objective that enforces the estimated Jacobian to be lower-triangular. To this end, we need to learn both a permutation $\pi$ for the causal ordering as well as a permutation $\pi_{\mathrm{ICA}}$ that resolves the indeterminacy in the noise variables introduced by ICA. We use the permuted absolute Jacobian $\mathbf{K}$ defined as

$$\mathbf{K} := \left| \mathbf{S}_{\mathrm{ICA}} \mathbf{J}_{\widehat{\boldsymbol{f}}^{-1}} \mathbf{S}_{\pi} \right| \tag{4}$$

where $\mathbf{S}_{\mathrm{ICA}}, \mathbf{S}_{\pi}$ are doubly-stochastic matrices that represent a soft permutation on both noise and observation variables, which we parametrize via Sinkhorn networks (Mena et al., 2018) and learn after ICA training—cf. § 5.1 and Fig. 10 for details. We then introduce a training loss inspired by LiNGAM (Shimizu et al., 2006) that encourages $\mathbf{K}$ to be lower-triangular by simultaneously maximizing i) the sum of the main diagonal and ii) the lower-triangular part, while also iii) minimizing the stricly-upper triangular part of $\mathbf{K}$,

$$\mathcal{L}_{\pi} = \sum_{i,j} \left[ \alpha_d \left( \mathbf{K} \right)^{-1}_{ii} - \alpha_l \left( \mathbf{K} \right)_{i \geq j} + \alpha_u \left( \mathbf{K} \right)_{i < j} \right], \tag{5}$$

where $i \in \{d, u, l\} : \alpha_i > 0$ The full learning algorithm is presented in Alg. 1.

Compared to LiNGAM, our method is differentiable and works for nonlinear SEMs; thus, it does not require iterating over all permutations. Although Sinkhorn networks (Mena et al., 2018) were previously proposed to represent permutation probabilities (Charpentier et al., 2022), we are the first to represent the indeterminacy of ICA with such models.

---

**Algorithm 1** Algorithm for multivariable CD and determining the causal order $\pi$

---

**Input:** dataset $D$, network parameters $\theta$, Sinkhorn networks $\mathbf{S}_{\text{ICA}}, \mathbf{S}_\pi$, contrastive loss $\mathcal{L}_{\text{CL}}$ (eq. (6)), ordering loss $\mathcal{L}_\pi$ (eq. (5)), positive scalars $\alpha_d, \alpha_u, \alpha_l$

Initialize $\theta$

**while** $\mathcal{L}_{\text{CL}}$ not converged **do**

    calculate $\mathcal{L}_{\text{CL}}$ for a batch from $D$

    update $\theta$

**end while**

extract $\mathbf{J}_{\widehat{f}^{-1}}$

**while** $\mathcal{L}_\pi$ not converged **do**

    $\mathbf{K} = \left| \mathbf{S}_{\text{ICA}} \mathbf{J}_{\widehat{f}^{-1}} \mathbf{S}_\pi \right|$

    $\mathcal{L}_\pi = \sum_{i,j} \left[ \alpha_d \left( \mathbf{K} \right)_{ii}^{-1} - \alpha_l \left( \mathbf{K} \right)_{i \geq j} + \alpha_u \left( \mathbf{K} \right)_{i < j} \right]$

    update $\mathbf{S}_{\text{ICA}}, \mathbf{S}_\pi$

**end while**

---

## 4 Identifiability in Contrastive Learning

There are fundamental limits to how much one can learn about a DGP from only i.i.d. observations: neither causal structure (Pearl, 2009), nor nonlinear mixing of independent signals (Hyvärinen & Pajunen, 1999) are identifiable in the general case. In this work, we describe a non–i.i.d. (contrastive) DGP, in which significantly more structure can be identified from observations.

In a contrastive DGP (Zimmermann et al. (2021); § 4.2), we generate so-called positive pairs containing two $d$–dimensional samples $(\boldsymbol{X}, \widetilde{\boldsymbol{X}})$. Underlying $\boldsymbol{X}$ and $\widetilde{\boldsymbol{X}}$ are a pair of latent variables of the same dimension $\boldsymbol{N}$ and $\widetilde{\boldsymbol{N}}$, such that $\boldsymbol{X} = \boldsymbol{f}(\boldsymbol{N})$ and $\widetilde{\boldsymbol{X}} = \boldsymbol{f}(\widetilde{\boldsymbol{N}})$. Both $\boldsymbol{N}$ and $\widetilde{\boldsymbol{N}}$ have statistically independent components, i.e. $\forall i, j : N_i \perp N_j$ and $\widetilde{N}_i \perp \widetilde{N}_j$. Furthermore, each component of $\widetilde{\boldsymbol{N}}$ depends only on the corresponding component of $\boldsymbol{N}$, such that $\forall i : \widetilde{N}_i \sim p(\cdot|N_i)$. In this work, we will often assume that the mapping $\boldsymbol{f}$ is defined as a SEM (§ 2).

### 4.1 Identifiability of causal graphs via the ICM principle

In the contrastive DGP, we draw i.i.d. samples of positive pairs. Inasmuch as we consider $\boldsymbol{X}$ and $\widetilde{\boldsymbol{X}}$ as two observations, the generative process is non–i.i.d.. This non–i.i.d. DGP leaves more fingerprints in the observed data, allowing the identification of causal dependencies that are non-identifiable in the i.i.d. case We will illustrate why this is the case in the following two-variable example.

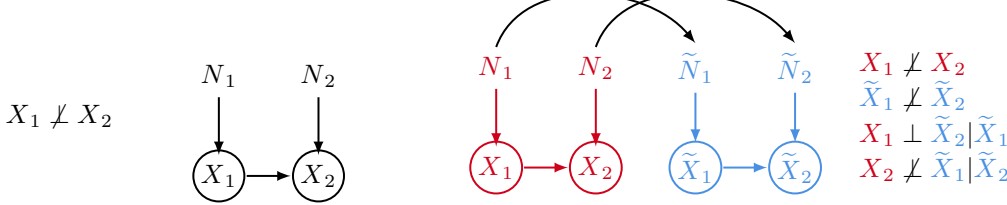

(a) No data augmentation.

(b) Data augmentation. Superscripts denote the two components of the positive pair.

Figure 2: Comparing conditional independencies between observables $\widetilde{X}_i$ in a bivariate model without (Fig. 2a) and with (Fig. 2b) data augmentations in the contrastive pair.

**Example 2** (Positive pairs induce additional conditional independencies)**.** *Assume a bivariate, faithful SEM without confounders, where $X_1 \to X_2$ (Fig. 2a). Observing only i.i.d. copies of $X_1$ and $X_2$, the direction of the cause–effect relationship cannot be discerned, the only statement we can make is $X_1 \not\perp X_2$ (Pearl, 2009).*

*However, consider observing positive pairs from the contrastive DGP, illustrated graphically in (Fig. 2b). As $\boldsymbol{X}$ and $\widetilde{\boldsymbol{X}}$ are dependent, there is a broader set of conditional independence statements we can make, and these resolve the ambiguity in the causal direction. Namely, $X_1$ (the cause component of $\boldsymbol{X}$) and $\widetilde{X}_2$ (the effect component of $\widetilde{\boldsymbol{X}}$) are statistically dependent via the path $X_1 \leftarrow N_1 \rightarrow \widetilde{N}_1 \rightarrow \widetilde{X}_1 \rightarrow \widetilde{X}_2$. However, conditioning on $\widetilde{X}_1$ blocks this path, thus $X_1 \perp \widetilde{X}_2 | \widetilde{X}_1$. Notably, such a conditional independence holds only when $X_1 \rightarrow X_2$, but would not hold, if the direction were reversed to $X_1 \leftarrow X_2$.*

Ex. 2 sheds light how Contrastive Learning (CL) enables CD by introducing additional conditional independencies in the positive pair. This line of reasoning connects our work to the Causal de Finetti (CdF) theorem (Guo et al., 2022), which proves identifiability of fully observed causal graphs under a very similar generative process. The key concept in the CdF is the notion of Independent Causal Mechanisms (ICM)[3]. That is, the assumption that various mechanisms that make up the generative process (e.g., individual equations in a SEM) change or vary in a statistically independent manner. In our generative model, when $\boldsymbol{f}$ is a SEM, the ICM principle manifests in the assumption that $\forall i \neq j : N_i \perp \widetilde{N}_j$. One can thus think of $\widetilde{\boldsymbol{X}}$ as an observed counterfactual outcome (also noted in Liu et al. (2023)), where the structural equations have been independently perturbed.

Exploiting the connection to CdF, one can show that in our generative process, all causal relationships become identifiable even in multivariable data in the absence of unobserved (confounder) variables. While showing this is somewhat involved in the asymmetric contrastive DGP of Zimmermann et al. (2021) presented here, other variants of contrastive DGPs—including the original model proposed in SimCLR (Chen et al., 2020) or (Dubois et al., 2022)—produce exchangeable positive pairs by relying on two augmented samples $(\widetilde{\boldsymbol{X}}^1, \widetilde{\boldsymbol{X}}^2)$, and thus map directly to the CdF setting.

Showing that causal relationships are identifiable from the conditional independence structure in CL is mathematically interesting, but does not yield a practical algorithm. As noted by Guo et al. (2022), the CdF requires an exponential number of conditional independence tests for multivariable CD. Here, we take a different approach, that exploits the full identifiability of the functional relationships $\boldsymbol{f}$ in the same setting.

### 4.2 Identifiability of the causal graph via identifiability of $\boldsymbol{f}$

We assume the setting of (Zimmermann et al., 2021, Thm. 6)—with the additional constraint that $\dim \boldsymbol{N} = \dim \boldsymbol{X} = d$—, under which, an inference model $\widehat{\boldsymbol{f}}^{-1}$ which minimizes a contrastive loss was proven to estimate the noise variables (often referred to as "sources" in the ICA literature) up to a composition of input independent permutations, sign flips, and rescaling. For completeness, we restate the assumptions both for the DGP (Assum. 2) in the main text and defer the model assumptions (Assum. F.1) to the appendix (Appx. F). We denote positive pairs with $\widetilde{(\cdot)}$ and negative pairs with $(\cdot)^-$.

**Assumption 2** (DGP on $\mathbb{R}^d$). *We assume that the DGP satisfies the following conditions:*
  *(i) The space of the noise variables to be a convex body (hyperrectangle); i.e., $\mathcal{N} \subseteq \mathbb{R}^d$;*
  *(ii) The observation space to be $\mathcal{X} \subseteq \mathbb{R}^d$;*
  *(iii) The generator (the SEM) $\boldsymbol{f}$ to*
      *1. be bijective,*
      *2. map $\mathcal{N} \subseteq \mathbb{R}^d \rightarrow \mathcal{X}$, and*
      *3. be differentiable in the vicinity of $\mathcal{N}$.*
  *(iv) The marginal distribution $p(\boldsymbol{N})$ over latent variables $\boldsymbol{N} \in \mathcal{N}$ is uniform[4]; i.e., $p(\boldsymbol{N}) = |\mathcal{N}|^{-1}$;*
  *(v) The conditional distribution over positive pairs $p(\widetilde{\boldsymbol{N}}|\boldsymbol{N})$ is a rotationally asymmetric generalized normal distribution (Subbotin, 1923) with a shape parameter $\alpha$ with the corresponding $L_\alpha$-metric (denoted as $\delta$), where $\alpha \geq 1 \wedge \alpha \neq 2$ [5]; i.e., $p(\widetilde{\boldsymbol{N}}|\boldsymbol{N}) = C_p^{-1}(\boldsymbol{N})e^{-\lambda\delta(\boldsymbol{N},\widetilde{\boldsymbol{N}})}$ with $C_p := \int e^{-\lambda\delta(\boldsymbol{N},\widetilde{\boldsymbol{N}})} d\widetilde{\boldsymbol{N}}$, where $\lambda > 0$ a parameter controlling the width of the distribution.*

---

[3]We note that Guo et al. (2022) develop CdF from the ICM principle; however, Pearl's autonomous mechanism principle Pearl (2009) might be a more appropriate term to use

[4]Since any random variable in $\mathbb{R}^d$ can be emulated by passing a uniformly distributed random variable through the corresponding inverse CDF, if the CDF is differentiable, we can abosrb it into $\boldsymbol{f}$

[5]In our experiments, we use $\alpha = 1$ (the Laplace distribution), since certain transitions in natural videos seem to follow the generalized Laplace distribution, and can be modeled successfully with the Laplace distribution (Klindt et al., 2021)

We parametrize the conditional distribution $q(\widetilde{\boldsymbol{N}}|\boldsymbol{N})$ via $\boldsymbol{f}^{-1}$ as in Assum. 2 and calculate the loss as:

$$\mathbb{E}_{\left(\boldsymbol{X},\widetilde{\boldsymbol{X}},\boldsymbol{X}^-\right)}\left[-\log\frac{\exp\left[-\delta\left(\widehat{\boldsymbol{f}}^{-1}(\boldsymbol{X}),\widehat{\boldsymbol{f}}^{-1}(\widetilde{\boldsymbol{X}})\right)/\boldsymbol{\tau}\right]}{\exp\left[-\delta\left(\widehat{\boldsymbol{f}}^{-1}(\boldsymbol{X}),\widehat{\boldsymbol{f}}^{-1}(\widetilde{\boldsymbol{X}})\right)/\boldsymbol{\tau}\right]+\sum_i^M\exp\left[-\delta\left(\widehat{\boldsymbol{f}}^{-1}(\boldsymbol{X}^-),\widehat{\boldsymbol{f}}^{-1}(\widetilde{\boldsymbol{X}})\right)/\boldsymbol{\tau}\right]}\right], \quad (6)$$

where $\widetilde{\boldsymbol{X}}$ is the positive pair, $\boldsymbol{X}^-$ are the negative pairs, and $M$ is the number of negative samples. During training one has access to observations $\boldsymbol{X}$, which are samples from these distributions transformed by the generator function (i.e., the SEM) $\boldsymbol{f}$.

**Compatibility of CL and CD.** Using observation pairs for CD might seem fundamentally different from conventional approaches, but there is a conceptual connection to interventions (Brouillard et al., 2020; Ke et al., 2020; Lippe et al., 2021; Mansouri et al., 2022; Bagi et al., 2023)—indeed, several methods rely on data pairs to identify the causal variables (Locatello et al., 2020; Brehmer et al., 2022; Von Kügelgen et al., 2021; Liu et al., 2023; Ahuja et al., 2022a). As stated by Liu et al. (2023), **the positive pairs in CL can be thought of as sparsity information about mechanism shifts, making the paradigm suitable for CD**. Locatello et al. (2020) relies on interventional pairs to for causal disentanglement. Brehmer et al. (2022) rely on pairs of pre-and post-interventional observations (with perfect interventions, which might be restrictive in practice (Liu et al., 2023)). Namely, since positive samples are "more similar" to the anchor point, the conditional distribution is more restricted (compared to the marginal; e.g., it should have a smaller variance) for the latent factors. However, this will only affect specific mechanisms (i.e., factors in the factorizing conditional), such as making the factor responsible for the class-determining variable degenerate (i.e., if the anchor depicts a chair, the mechanism encoding the class information will have a delta distribution). Ahuja et al. (2022a) provides identifiability results for sparse perturbations (generalizing (Von Kügelgen et al., 2021); thus, emphasizing the connection to between interventions and contrastive methods. The recent work of Bagi et al. (2023) proposes a variational inference-based approach from interventional data, where the authors partition their latent space into invariant (content) and variant (style) features, which is a paradigm also found in CL, including identifiability guarantees and competitive performance on the Causal3DIdent dataset (Von Kügelgen et al., 2021). Compared to assuming perfect interventions (Brehmer et al., 2022), degenerate (delta) conditionals for the invariant (content) partition of the latent space (Von Kügelgen et al., 2021), or Gaussian/Gaussian Mixture priors (Bagi et al., 2023), our rotationally asymmetric generalized normal assumption on the conditional (Assum. 2(v)) is less restrictive.

**Differences between identifying the DAG and $\boldsymbol{f}$.** Assum. 2 implies restrictions on the class of SEM we can represent in this framework. In particular, Assum. 2(iv) requires that the noise variables are uniform. This however, is a minimal restriction, considering that any real-valued random variable can be emulated by passing a uniform random variable through its inverse cumulative distribution function (CDF). Thus, so long as the inverse CDF is differentiable, we can absorb it into $\boldsymbol{f}$ without modifying the causal structure implied by the SEM. Assum. 2(v) relates to the type of random perturbation under which the positive pairs are generated. We note that Assum. 2 is specific to the contrastive ICA framework of (Zimmermann et al., 2021) but our approach is not fundamentally limited to the this setting, and can be used in principle in any situation where nonlinear ICA is identifiable, such as in (Hyvärinen & Morioka, 2016; 2017; Morioka et al., 2021; Monti et al., 2019).

That is, our results state that identifiability of $\boldsymbol{f}$ entails identifiability of the causal graph. However, this is not necessarily true in the other direction, since knowing the Jacobian (which is sufficient to recover the DAG) does not contain all information about $\boldsymbol{f}$. It remains an open question how practically relevant these differences are.

## 5 Experiments

### 5.1 Experimental setup

**Data Generating Process (DGP).** We experiment with three DGPs: i) linear and ii) nonlinear SEMs (in the form of $\boldsymbol{X} = \boldsymbol{f}(\mathbf{W}\boldsymbol{N})$), as well as with iii) Multi-Layer Perceptrons (MLPs) with triangular weight matrices (as used in (Monti et al., 2019)). In all cases, the nonlinear activations (i.e., $\boldsymbol{f}$) are leaky ReLUs (with a slope of 0.25 for the SEMs and 0.1 for the triangular MLPs). For the SEM DGPs, we exlore three

Table 1: Validation of Lem. 1 for linear and nonlinear SEMs with **unknown causal ordering** to measure how well our method recovers the causal ordering. Mean Correlation Coefficient (MCC) measures identifiability, $|\mathcal{E}^*|$ is the maximum number of edges in a DAG, $\text{Acc}_\pi$ is the accuracy of recovering the pairwise causal ordering $\pi$, whereas $\pi$ gives the ratio of learning a (any) permutation in $\mathbf{S}_\pi$ and SHD is the Structural Hamming Distance

| | | LINEAR | | | | | NONLINEAR | | | | |
|---|---|---|---|---|---|---|---|---|---|---|---|
| $|\mathcal{E}^*|$ | $d$ | MCC | Acc | $\text{Acc}_\pi$ | $\pi$ | SHD | MCC | Acc | $\text{Acc}_\pi$ | $\pi$ | SHD |
| 6 | 3 | 1. | 1. | 1. | 1. | 0. | 1. | 1. | 1. | 1. | 0. |
| 15 | 5 | $0.989_{\pm 0.039}$ | $0.998_{\pm 0.009}$ | $0.974_{\pm 0.078}$ | 0.76 | $0.002_{\pm 0.009}$ | $0.988_{\pm 0.039}$ | $0.994_{\pm 0.021}$ | $0.957_{\pm 0.129}$ | 0.583 | $0.006_{\pm 0.021}$ |
| 36 | 8 | $0.834_{\pm 0.238}$ | $0.935_{\pm 0.081}$ | $0.851_{\pm 0.183}$ | 0.414 | $0.065_{\pm 0.081}$ | $0.781_{\pm 0.219}$ | $0.934_{\pm 0.051}$ | $0.889_{\pm 0.15}$ | 0.345 | $0.066_{\pm 0.051}$ |
| 55 | 10 | $0.852_{\pm 0.251}$ | $0.931_{\pm 0.051}$ | $0.921_{\pm 0.147}$ | 0.233 | $0.069_{\pm 0.051}$ | $0.794_{\pm 0.255}$ | $0.924_{\pm 0.073}$ | $0.739_{\pm 0.252}$ | 0.276 | $0.076_{\pm 0.073}$ |

options: a) no permutation w.r.t. the causal ordering (i.e., only the ICA permutation remains); b) a sparse DGP (with each $X_i - X_j$ edge being nonzero with a 0.25 probability); and c) permuted causal ordering (with dense $\mathcal{A}$). Additionally, we ensure that the ordering of $N_i$ is unique (all cases), and that the DGP weights are $\gg 0$ (for the SEM DGPs) as otherwise we would be unable to distinguish weak connections from small elements in the Jacobian. That is, the estimate of a weak connection could be the same order of magnitude as the estimate of a zero element due to the stochasticity of training—we do not enforce this property for the triangular MLPs to compare to the results of (Monti et al., 2019), where such modification was not present. For the *permuted* SEM DGPs, we sample 6 different orderings and 5 seeds for each problem dimensionality $\{3; 5; 8; 10\}$—the number of seeds is 10 for non-permuted and sparse SEMs. For the triangular MLP, we use $d = 6$ and 5 seeds to compare to (Monti et al., 2019, Fig. 2) and vary the number of layers in the mixing. To use contrastive NLICA for training the inference model, the DGPs needs to satisfy the assumptions underlying the proof of identifiability (Zimmermann et al., 2021, Thm. 6)): the latent space is a hyperrectangle in $\mathbb{R}^d$, the marginal $p(\boldsymbol{N})$ is uniform, the conditional $p(\widetilde{\boldsymbol{N}}|\boldsymbol{N})$ is Laplace, $\boldsymbol{X}$ is generated by a smooth and bijective mapping.

**Inference model.** To (strongly) identify the SEM, we use contrastive NLICA (Zimmermann et al., 2021)—which is consistent when the number of negative samples goes to infinity—to estimate $\widehat{\boldsymbol{f}}^{-1}$ with a hyperrectangle latent space in $\mathbb{R}^d$ and the contrastive loss uses the same metric as the conditional, which is $L_1$ for our case (Assums. 2 and F.1). Our architecture for the inference model is the same MLP as in (Zimmermann et al., 2021) (Tab. 6). To account for the permutation indeterminacies, we use two Sinkhorn networks (Mena et al., 2018) (similar to Charpentier et al. (2022)). A Sinkhorn network is a trainable parametrization of soft-permutation matrices (the Birkhoff polytope) (Mena et al., 2018), consisting of two levels: i) the Sinkhorn operator (Fig. 10) normalizes each row and column (in this order) of a matrix to one, relyin on the log-sum-exp operator; ii) the network layer contains the trainable

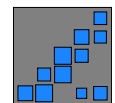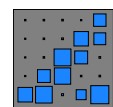

Figure 3: Hinton diagrams ($d = 5$): ground truth (left), estimate (right). Size equals magnitude.

matrix $\mathbf{W}$ with the scalar temperature value $\boldsymbol{\tau}$ to ensure convergence to the Birkhoff polytope's vertices, i.e., to yield a (hard) permutation matrix. We observed that setting the lowest $d(d - 1)/2$ elements (for dense DAGs) to zero and converting the resulting $\mathbf{K}$ matrix to binary often helped the convergence of the Sinkhorn networks. We calculate the Jacobian of the inference model with the `autograd` module of PyTorch (Paszke et al., 2019) in the forward pass and vectorize the operation with the recently released `functorch` library (Horace He, 2021). Moreover, instead using max to aggregate the different Jacobians over the batch, we found using the mean operator more stable in practice.

**Metrics.** We measure learning the correct ordering by the ordering accuracy ($\text{Acc}_\pi$, only for the permuted case)—i.e., the ratio of causal variable pairs $\forall i < j : (N_i, N_j)$, such that the ranking (expressed by $\text{sign}(i - j)$) matches that in the inferred (permuted) ordering $\pi$, i.e., $\text{sign}(\pi(i) - \pi(j))$. To normalize, we divide by the number of distinct edge pairs ($1/2(d-1)d$) We also report the accuracy (Acc, i.e., the ratio of correctly identified edges, or lack thereof, divided by $|\mathcal{E}^*|$) and the Structural Hamming Distance (SHD) (we use

Table 2: Causal discovery performance for linear and nonlinear SEMs with **known causal ordering**. Mean Correlation Coefficient (MCC) measures identifiability, $|\mathcal{E}^*|$ is the maximum number of edges in a DAG, Acc is accuracy, Ours is our proposal, HSIC refers to using HSIC independence tests, and SHD is the Structural Hamming Distance

| $|\mathcal{E}^*|$ | $d$ | LINEAR | | | | NONLINEAR | | | |
|---|---|---|---|---|---|---|---|---|---|
| | | MCC | Acc(Ours) | Acc(HSIC) | SHD | MCC | Acc(Ours) | Acc(HSIC) | SHD |
| 6 | 3 | 1. | 1. | $0.7_{\pm 0.1}$ | 0. | 1. | 1. | $0.741_{\pm 0.105}$ | $0.049_{\pm 0.14}$ |
| 15 | 5 | $0.969_{\pm 0.066}$ | $0.928_{\pm 0.131}$ | $0.828_{\pm 0.116}$ | $0.072_{\pm 0.131}$ | $0.94_{\pm 0.09}$ | $0.858_{\pm 0.172}$ | $0.8_{\pm 0.102}$ | $0.142_{\pm 0.171}$ |
| 36 | 8 | 1. | 1. | $0.682_{\pm 0.17}$ | 0. | $0.982_{\pm 0.029}$ | $0.872_{\pm 0.198}$ | $0.823_{\pm 0.142}$ | $0.128_{\pm 0.198}$ |
| 55 | 10 | $0.965_{\pm 0.03}$ | $0.832_{\pm 0.176}$ | $0.551_{\pm 0.003}$ | $0.168_{\pm 0.176}$ | $0.962_{\pm 0.025}$ | $0.636_{\pm 0.239}$ | $0.638_{\pm 0.134}$ | $0.364_{\pm 0.239}$ |

$1e-3$ as the threshold in all scenarios) for inferring the edges of the DAG, as is standard practice in the literature (Lachapelle et al., 2020; Monti et al., 2019; Ke et al., 2020; Vowels et al., 2022).

**Comparison.** We use the linear and nonlinear SEM DGPs to showcase that our method can infer the DAG while also learning the correct ordering. Then, we compare to NonSENS (Monti et al., 2019), which, unlike our proposal, does CD on an edge-by-edge basis. Thus, the causal ordering $\pi$ does not affect how NonSENS operates. We use the HSIC independence test (Gretton et al., 2005) on top of contrastive NLICA (Zimmermann et al., 2021) to provide a close comparison with NonSENS (Monti et al., 2019). Notably, since our assumptions provide identifiability up to generalized permutations, there is no need to perform linear ICA on top of contrastive NLICA. Thus, we test independence between the observations $X_i$ and the *inferred* noise variables $\hat{N}_j$—although the number of tests is $d^2$, we use a Bonferroni correction factor of 4, since each edge is determined based on four tests (Monti et al., 2019).

### 5.2 Results

In all experiments except those in Tab. 1, we used the output of the matching problem as an oracle (solved via the Hungarian algorithm (Kuhn, 1955)) to correct for the permutation indeterminacy of ICA.

**The permutation indeterminacies can be resolved (verifying Lem. 1).** Tab. 1 corroborates the result of Lem. 1: it is possible to resolve the permutation indeterminacy by assuming a DAG DGP. However, $\text{Acc}_\pi$ strongly depends on the performance of NLICA, measured by Mean Correlation Coefficient (MCC). As MCC deteriorates, the correct causal ordering cannot be recovered. Nonetheless, erroneous solutions resulting from training stochasticity (the most frequent problem according to our observations) can be simply filtered out: in this case the doubly stochastic matrices usually do not converge to a permutation matrix. Inspecting their elements or automatically rejecting such solutions is straightforward. Thus, we report two quantities in Tab. 1: $\text{Acc}_\pi$ is the ratio of inferring the order of causal variable pairs *when the Sinkhorn networks converged to permutation matrices*; $\pi$ (with a slight abuse of notation), on the other hand, reports the ratio of the successful

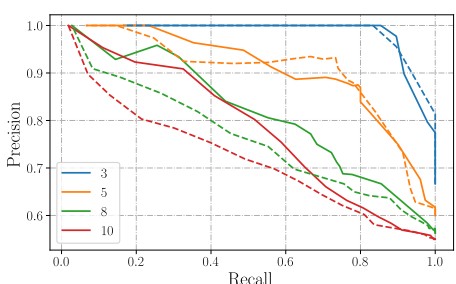

Figure 4: Precision vs recall for thresholds in $[1e-7; 1e0]$ for linear (dashed) and nonlinear (solid) *sparse* SEMs

attempts to recover permutation matrices. Clearly, failing to converge to a permutation matrix is the bottleneck of this step, since despite failing to scalably recover $\pi$, in case of converging to a permutation matrix the captured graph reflects most of the edges. This is reported by the $\text{Acc}_\pi$ column that is calculated after applying the learned (not necessarily correct) permutations.

**Competitive performance on linear and nonlinear SEMs.** Tab. 2 demonstrates (with $\pi$ being known) that our method outperforms HSIC in the linear case and is at least comparable to HSIC in the nonlinear case—note that the entries in $\mathbf{J}_{\hat{f}^{-1}}$ were ordered by absolute value and the smallest ones were zeroed out—namely, these are the elements of the Jacobian that most probably correspond to the zeros in the true Jacobian. However, this modification might require additional knowledge about the sparsity of the

DAG. Fig. 4 describes how precision and recall changes for threshold values from $[1e{-}7; 1e0]$ for *sparse* DAGs. Notably, the nonlinear curves are better than the linear ones. For additional results on sparse SEMs (Tab. 7) and SEMs with unknown causal ordering (Tab. 8, evaluation is done after accounting for the causal ordering), cf. Appx. G. For sparse SEMs, HSIC is slightly better for larger graphs, whereas in the case of unknown causal ordering, our proposal has better accuracy in most cases. To visualize the inferred graph structure, we plot a Hinton diagram of the true and estimated Jacobians in Fig. 3, showing that $\mathbf{J}_{\widehat{f}^{-1}}$ can capture the edges of an underlying *sparse* DAG.

**Competitive performance on triangular MLPs from (Monti et al., 2019).** Tab. 3 summarizes our results with the triangular MLP of (Monti et al., 2019). Despite having small weights in the ground truth Jacobian $\mathbf{J}_{f^{-1}}$ (appr. $2e{-}3$ for one and $1e{-}8$ for five layers), our method was able to infer most edges in the DAG. Importantly, the resulting accuracies are larger than those of our adapted version of NonSENS (Monti et al., 2019). Moreover, our method has the advantage of simultaneously inferring all edges based on the structure of $\mathbf{J}_{\widehat{f}^{-1}}$—thus, it does not require $d^2$ pairwise independence test for a DAG with $d$ nodes. Our application of HSIC independence tests resulted in surprisingly low accuracy, despite utilizing an NLICA method with identifiability guarantees up to generalized permutations. Interestingly, HSIC resulted in (close-to) chance-level

Table 3: Causal discovery performance for the triangular MLP from (Monti et al., 2019) with $d = 6$. $|l|$ denotes the number of MLP layers, Acc the accuracy, Ours is our proposal, HSIC refers to using HSIC independence tests. Chance level is (for the dense MLP) $21/36 = 0.583$

| $|l|$ | MCC | Acc (Ours) | Acc (HSIC) |
|---|---|---|---|
| 1 | 1. | $0.933_{\pm 0.042}$ | 0.583 |
| 2 | 1. | 0.944 | 0.583 |
| 3 | $0.997_{\pm 0.003}$ | 1. | 0.583 |
| 4 | $0.978_{\pm 0.016}$ | $0.922_{\pm 0.097}$ | $0.6_{\pm 0.033}$ |
| 5 | $0.603_{\pm 0.062}$ | $0.711_{\pm 0.054}$ | $0.589_{\pm 0.011}$ |

performance in our repeated experiments—by careful inspection of the DGP, we found that the weights are in the order of $1e{-}4$, which might explain such bad performance. As noted above, since Monti et al. (2019) did not constrain the weights, we used a uniform initialization scheme, which might led to mismatching experimental conditions. Though the use of HSIC was inspired by NonSENS (Monti et al., 2019), since we made different assumptions on the DGP, the results only represent HSIC's (but not NonSENS's) performance.

## 6 Related work

We provide a detailed comparison of related CD methods (Tab. 4) and the use of the Jacobian (Tab. 5) in Appx. D.

**Independence tests for CD.** Traditional CD methods (Pearl, 2009; Spirtes & Zhang, 2016; Spirtes et al., 2000; Peters et al., 2017) rely on statistical (conditional) independence tests to infer the graph structure. Recent works also leverage such tests (Shimizu et al., 2006; Monti et al., 2019; Guo et al., 2022; Karlsson & Krijthe, 2022) to uncover hidden confounders (Karlsson & Krijthe, 2022), for bivariate (Janzing et al., 2009; Monti et al., 2019) or multivariable CD (Guo et al., 2022) for nonlinear SEMs. LiNGAM (Shimizu et al., 2006), which inspired our work, also relies on independence tests to prune edges. Although independence tests provide additional information via significance values, they are not differentiable and can be costly, as $d$ latents require $d^2$ tests.

**Optimization-based CD.** Zheng et al. (2018) introduced the continuous optimization-based NOTEARS algorithm for linear SEMs, which has inspired further research (Khemakhem et al., 2021; Lorch et al., 2021; Ng et al., 2022; Schölkopf et al., 2021a; Yu et al., 2019; Lachapelle et al., 2020; Kalainathan et al., 2018) to provide differentiable methods for CD in neural networks. Most of the differentiable solutions (Khemakhem et al., 2021; Ng et al., 2022; Schölkopf et al., 2021a; Yu et al., 2019) constrain the function class, some of them (Lachapelle et al., 2020; Kalainathan et al., 2018) both the function class and the data distribution.

**Using the adjacency matrix $\mathcal{A}$.** Our work shows that the adjacency matrix $\mathcal{A}$ and the Jacobian of the inference model $\mathbf{J}_{\widehat{f}^{-1}}$ can both be used to model the edges in a graph. We review both, starting with the adjacency matrix for CD: Zheng et al. (2018) use $\mathcal{A}$ as a regularizer in NOTEARS, Ng et al. (2022) reformulates the SEM with an adjacency matrix for additive models, Schölkopf et al. (2021a) models $\mathcal{A}$ with an LSTM in a

variational framework. In (Brouillard et al., 2020), $\mathcal{A}$ appears for the interventional case. Lorch et al. (2022) leverage amortized variational inference for CD, where they deploy multi-head attention Vaswani et al. (2017) and use the softmax probabilities as a proxy for the adjacency matrix (i.e., their model represents the probability of edges in the graph). Charpentier et al. (2022) defines a probabilistic model over $\mathcal{A}$ to differentiably sample DAGs, then use variational inference to estimate the causal structure, similar to Faria et al. (2022). The proposed method has strong empirical performance, but does not provide theoretical guarantees for CD.

**Using the Jacobian.** The Jacobian matrix of either the generative ($\mathcal{N} \rightarrow \mathcal{X}$) or the inference ($\mathcal{X} \rightarrow \mathcal{N}$) models are used throughout the literature, both for identifiability and CD (Tab. 5). LiNGAM Shimizu et al. (2006) uses the Jacobian (i.e., a constant matrix) to infer the DAG in the linear case, whereas Lachapelle et al. (2020) calculates the Jacobian of the inference network to enforce acyclicity, generalizing to nonlinear additive models. Rolland et al. (2022) consider the same model class as Lachapelle et al. (2020), but they rely on the Jacobian of the score function. Leveraging properties of the Jacobian is also present in the identifiability literature: Independent Mechanism Analysis (IMA) relies on the assumption that the generative model's Jacobian has orthogonal columns Gresele et al. (2021)—our work reasons about the inference network's Jacobian, without functional constraints. Although the inspiration comes from the causal principle of independent mechanisms, the claims are about identifiability and not about CD: the IMA function class is locally identifiable, whereas the subclass of conformal maps are identifiable Buchholz et al. (2022). Similar to IMA, Zheng et al. (2022) also utilize the Jacobian of the generative model and prove identifiability for NLICA under a sparsity assumption. Atanackovic et al. (2023) propose a Bayesian approach for CD in dynamical systems, including cyclic graphs, where they associate the graph's edges with the sparsity pattern of the Jacobian of the SEM (in this case an ODE), but the authors do not prove identifiability. That is, although the use of the Jacobian is prevalent in the literature, to the best of our knowledge, we are the first to use the Jacobian of the inference model for causal models without constraining the function class (but using non–i.i.d. data, while providing identifiability guarantees.

**CD from interventions.** Many algorithms can incorporate interventions (Brouillard et al., 2020; Ke et al., 2020; Lippe et al., 2021; 2022; Lorch et al., 2021). Interestingly, (Ke et al., 2020) provide an extension of (Yu et al., 2019; Zheng et al., 2018) to interventional data, and of the bivariate method of (Bengio et al., 2020) to a multivariable one. It is remarkably similar to our proposal, as both make assumptions only on the data (i.e., admitting general nonlinear functional relationships), but as (Ke et al., 2020) requires interventions, its path is orthogonal to ours. So is the work of (Lippe et al., 2021), which removes any requirement on the data, scales to multiple variables, but requires interventions.

## 7 Discussion

**Limitations.** Our theory requires the guarantees of strong identifiability but not the use of a specific (NLICA) algorithm. Though our experiments demonstrate that fulfilling strong identifiability is sufficient for CD, we do not vary the NLICA algorithm. Additionally, we acknowledge that since contrastive NLICA requires unique assumptions via the positive pair, it is non-trivial to design a task where the assumptions for multiple methods hold, making comparisons challenging. Our method's applicability is limited for inferring weak edges, similar to (Shimizu et al., 2006; Tashiro et al., 2014; Shahbazinia et al., 2021; Lachapelle et al., 2020). As demonstrated in § 5, despite its competitive performance, the success of our proposed method highly relies on the performance of NLICA, which can be limited for higher-dimensional problems. Nonetheless, based on our comparisons, this seems to be an issue for the HSIC independence test as well. A possible explanation could be that the class of SEMs is harder to learn with specific NLICA algorithms; indeed, we observed that deploying contrastive NLICA (Zimmermann et al., 2021) achieves much better MCC on general (non-triangular) invertible MLPs. To ensure that particular entries in the Jacobian are non-zero everywhere, our assumptions require that the underlying DAG for the DGP is the same for all data points, which might be restrictive . For instance, if the DAG models the interaction of physical objects, then cause-effect relationships are only present when, e.g., the objects are touching each other or their magnetic/electric fields affect each other—in the literature, this setting is considered in (Sontakke et al., 2021; Seitzer et al., 2021).

**CD with identifiability beyond CL.** As noted in the Limitations section above, our method is agnostic to how we achieve identifiability; however, our investigation only showcased contrastive NLICA. To emphasize that other NLICA–based methods are applicable for CD, we discuss how Time-Contrastive Learning

(TCL) Hyvärinen & Morioka (2016) and the Sparse Mechanism Shift (SMS) hypothesis (Schölkopf et al., 2021b) can be leveraged for CD. Monti et al. (2019) relies on TCL for bivariate CD. The authors show the correspondence between SEMs and the ICA generative model and study temporal sequences. Since the arrow of time defines a cause-effect relationship, relying on TCL is compatible with CD. The assumptions for identifiability in TCL (Hyvärinen & Morioka, 2016, Thm. 1) require smooth invertible functions, $\dim \mathbf{N} = \dim \mathbf{X}$, and exponential family distributions with *sufficient variability.* That is, TCL needs access to temporal data from different segments, where the intervention targets are the variance parameters. Perry et al. (2022) rely on the SMS hypothesis (Schölkopf et al., 2021b) to provably identify causal structures. Assuming that only a subset of mechanisms changes in each environment, the setting is akin to sufficient variability across time segments in TCL.

**Unknown causal ordering.**  Accounting for the causal ordering is, to the best of our knowledge, only found in (Shimizu et al., 2006). Binary CD methods such as (Monti et al., 2019) alleviate this step as they work on an edge-by-edge basis. Other non-ICA-based methods can also avoid this step since the DAG is *invariant* to changes in the causal ordering—meaning that reordering $X_i$ in the observation vector $\mathbf{X}$ (cf. Defn. A.11) does not affect the edges of the graph, only their representation in form of an adjacency matrix. However, to resolve the permutation indeterminacy of ICA, we need to account for the causal ordering, since only then can the Jacobian be lower-triangular. Although extracting a lower-triangular Jacobian is easier to interpret and potentially better suited, e.g., as a building block of causal representation learning (since the causal ordering of $N_i$ is always the same), our method extracts the DAG even without resolving these indeterminacies. That is, our demonstration that the permutation indeterminacies can be resolved should mostly be considered as corroboration of Lem. 1.

**Extensions to related work.**  Using neural networks for CD is discussed in several papers (Monti et al., 2019; Khemakhem et al., 2021; Lachapelle et al., 2020; Lippe et al., 2021; 2022; Brouillard et al., 2020), many of them uses the adjacency matrix, the Jacobian of the inference network (Shimizu et al., 2006; Schölkopf et al., 2021a; Lachapelle et al., 2020) or that of the score function Rolland et al. (2022). On the other hand, the Jacobian of the generative model is prevalent in the identifiability literature Gresele et al. (2021); Buchholz et al. (2022); Zheng et al. (2022), but they do not make claims about CD. Furthermore, methods that can handle general nonlinear relationships either require interventions (Brouillard et al., 2020; Lippe et al., 2021; 2022) or rely on independence tests (Guo et al., 2022; Monti et al., 2019). Our method was inspired by LiNGAM (Shimizu et al., 2006) to use the Jacobian of the inference network for inferring the DAG and utilizes NLICA (similar to Monti et al. (2019)) to provide theoretical guarantees (Props. 1 and 2) for multivariable CD. Furthermore, we also prove (Lem. 1) and demonstrate (Tab. 1) that the permutation indeterminacy of ICA—and that of an unknown causal ordering—can be resolved in the nonlinear case. Concurrent to our work, Morioka & Hyvarinen (2023) leverage CL for multimodal data and show that under specific assumptions both identifiability of the latent factors and CD are possible.

**Conclusion.**  Our method uses the Jacobian of the inference function (mapping from observables to independent variables) and can be thought as a generalization of LiNGAM (Shimizu et al., 2006) to nonlinear Causal Discovery (CD). We prove that the inverse DGP's Jacobian captures the sparsity structure of the DAG (Prop. 1), and show that under strong identifiabilty, the inference model also encodes the same information (Prop. 2). For the latter, we leverage that causal models enable resolving the permutation indeterminacy of ICA under certain assumptions (Lem. 1). We introduced a two-step process to leverage strong identifiability for inferring the DAG of multivariable causal models without constraints on the function class, but assuming non–i.i.d. data. That is, our approach leverages NLICA with auxiliary information (coming from the positive pairs, cf. Ex. 2) for CD. *We do not claim that NLICA is a CD method per se; however, we show that when the underlying generative model can be described by a causal graph and we have non–i.i.d. data, then CD is possible with NLICA.* Particularly, we show that contrastive NLICA (Zimmermann et al., 2021) is compatible with CD. Although the use of the Jacobian is prevalent in the literature, to the best of our knowledge, **we are the first to use the inference model's Jacobian for causal discovery without constraining the function class (but using non–i.i.d. data), while also providing identifiability guarantees**. Since we do not use conditional independence tests, but learn the causal ordering with Sinkhorn networks, our method provides an end-to-end solution for CD and avoids the cost of exponentially many independence tests. We experimentally demonstrate that our proposal can infer the DAG in multiple synthetic data sets.

**Author Contributions**

Wieland Brendel initiated the project and guided it together with Ferenc Huszár, with input from Matthias Bethge and Bernhard Schölkopf. Ferenc Huszár and Patrik Reizinger derived the theory, Yash Sharma set up the codebase from (Zimmermann et al., 2021), Patrik Reizinger ran the experiments, evaluated the results, and created the figures. All authors wrote the paper.

**Acknowledgments**

The authors would like to thank Ricardo Pio Monti and Scott W. Linderman for helpful correspondence. We thank Luigi Gresele and Julius von Kügelgen for fruitful discussions. Wieland Brendel acknowledges financial support via an Emmy Noether Grant funded by the German Research Foundation (DFG) under grant no. BR 6382/1-1. Wieland Brendel is a member of the Machine Learning Cluster of Excellence, EXC number 2064/1 – Project number 390727645. Ferenc Huszár acknowledges financial support from UKRI, this research was partially supported by a Turing AI Fellowship, grant ref: EP/W002965/1. The authors thank the International Max Planck Research School for Intelligent Systems (IMPRS-IS) for supporting Yash Sharma and Patrik Reizinger. Patrik Reizinger acknowledges his membership in the European Laboratory for Learning and Intelligent Systems (ELLIS) PhD program.

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

# Appendix

## A    SEMs

**Definition A.1** (SEM)**.** *A SEM describes causal relationships via a set of structural assignments (Peters et al., 2017):*

$$X_i := f_i\left(\boldsymbol{Pa}_i, N_i\right), \qquad \forall i \in \mathcal{I} = \{1, \ldots, d\},\tag{7}$$

*where $X_i$ are the endogenous, $N_i$ the exogenous/noise variables, $\boldsymbol{Pa}_i \subseteq \boldsymbol{X} \setminus \{X_i\}$ denotes the parent set of $X_i$, $\mathcal{I}$ the set of indices, and $f_i$ the mappings.*

**Definition A.2** (Reduced form of SEM)**.** *The reduced form of the SEM expresses all $X_i$ as a function of only the $N_i$ variables, i.e.:*

$$X_i := f_i\left(\boldsymbol{N}^i\right), \qquad \forall i \in \mathcal{I} = \{0, \ldots, d-1\},\tag{8}$$

*with the same notation as in Defn. A.1, slightly abusing $f_i$ and denoting a subset of $\boldsymbol{N}$ by $\boldsymbol{N}^i \subseteq \boldsymbol{N}$.*

**Definition A.3** (Chain)**.** *A graphical structure of three nodes $X_k$, $X_p$, $X_q$ is called a chain if two nodes are both parents of the third. Graphically, this means:*

Figure 5: Visualization of a chain. Conditioning on the middle node (denoted with gray color) blocks the path $X_p \rightarrow X_k \rightarrow X_q$.

*That is, the following conditional independence (denoted by $\perp$) relationship holds:*

$$X_p \perp X_q | X_k\tag{9}$$

**Definition A.4** (Collider)**.** *A graphical structure of three nodes $X_k$, $X_p$, $X_q$ is called a collider if two nodes are both parents of the third. Graphically, this means:*

Figure 6: Visualization of a collider. Conditioning on the collider node (denoted with gray color) opens the path $X_p \rightarrow X_k \leftarrow X_q$.

*That is, the following conditional dependence (denoted by $\not\perp$) relationship holds:*

$$X_p \not\perp X_q | X_k\tag{10}$$

**Definition A.5** (Fork)**.** *A graphical structure of three nodes $X_k$, $X_p$, $X_q$ is called a fork if one node is the parent of the two other nodes. Graphically, this means:*

Figure 7: Visualization of a fork. Conditioning on the fork node (denoted with gray color) blocks the path $X_p \leftarrow X_k \rightarrow X_q$.

*That is, the following conditional independence (denoted by $\perp$) relationship holds:*

$$X_p \perp X_q | X_k\tag{11}$$

**Definition A.6** (Confounder (unobserved common cause)). *In a DAG with nodes $X_i : \forall i \in \mathcal{I} = \{1, \ldots, d\}$ a node $X_k$ is called a confounder if there exist at least two $p, q \in \mathcal{I} : X_k \in \boldsymbol{Pa}_p \wedge X_k \in \boldsymbol{Pa}_q$ and $X_k$ is unobserved. Graphically,*

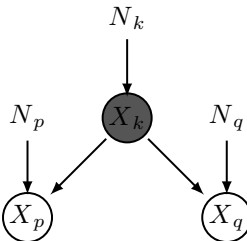

Figure 8: Visualization of a confounder (unobserved common cause), indicated by a gray node color.

**Definition A.7** (Causal ordering). *The causal ordering $\pi$ is a bijective automorphism on the index set $\mathcal{I}$. Namely, $\pi : \mathcal{I} \to \mathcal{I}$ so that $\forall X_i \neq X_j$, it holds that if $\pi(i) < \pi(j) \implies X_j \notin \boldsymbol{Pa}_i$.*

The definition means that only a node with a smaller index in $\pi$ can be a parent of a node with a larger index. Note that though $X_i$ *can* be a parent of $X_j$, it is not necessary, but $X_j$ *cannot* be a parent of $X_i$. Multiple orderings may exist, e.g. if there are multiple $X_i$ so that they only have a single parent. $\pi$ helps to have a unique description of the edges in the graph. Namely, if the edges are organized in the adjacency matrix $\boldsymbol{\mathcal{A}}$ according to $\pi$, then $\boldsymbol{\mathcal{A}}$ will be strictly lower triangular.

**Definition A.8** (Adjacency matrix). *The adjacency matrix $\boldsymbol{\mathcal{A}}$ is a binary $d \times d$ matrix, where $\boldsymbol{\mathcal{A}}_{ij} = 1 \iff X_j \in \boldsymbol{Pa}_i$. The rows of $\boldsymbol{\mathcal{A}}$ are ordered by $\pi$; thus, $\boldsymbol{\mathcal{A}}$ is strictly lower-triangular.*

$\boldsymbol{\mathcal{A}}$ only describes the edges of the DAG, which gives the direct cause-effect relationships. Nodes can be influence each other via paths (i.e., a set of directed edges that can be traversed between the two nodes), which can be described by the connectivity matrix $\boldsymbol{\mathcal{C}}$

**Definition A.9** (Connectivity matrix). *The connectivity matrix $\boldsymbol{\mathcal{C}}$ is a binary $d \times d$ matrix, where $\boldsymbol{\mathcal{C}} = 1 \iff \exists p : X_j \to \cdots \to X_i$. $\boldsymbol{\mathcal{C}} = \sum_{k=1}^{d} \boldsymbol{\mathcal{A}}^k$. The rows of $\boldsymbol{\mathcal{C}}$ are ordered by $\pi$; thus, $\boldsymbol{\mathcal{C}}$ is strictly lower-triangular.*

**Assumption A.1** (Structural faithfulness). *The set of $\boldsymbol{N}$'s that induces additional zeroes (i.e., a sparser DAG) in the Jacobians $\mathbf{J}_{\boldsymbol{f}}$, $\mathbf{J}_{\boldsymbol{f}^{-1}}$ has zero measure, i.e., both Jacobians describe the sparsity structure of the underlying DAG DGP with probability one ($\mathbf{J}_{\boldsymbol{f}}$ w.r.t. $\boldsymbol{\mathcal{C}}$, as shown in Lem. A.1; $\mathbf{J}_{\boldsymbol{f}^{-1}}$ w.r.t. $\boldsymbol{\mathcal{A}}$). Alternatively, the structural independencies are reflected in a functional form via $\mathbf{J}_{\boldsymbol{f}}/\mathbf{J}_{\boldsymbol{f}^{-1}}$. We call this property structural faithfulness.*

**Definition A.10** (DGP with known $\pi$). *The DGP is described by the SEM, when $\pi$ is known. I.e., the flow of information is: $\boldsymbol{N} \xrightarrow{SEM} \boldsymbol{X}$.*

**Definition A.11** (DGP with unknown $\pi$). *The DGP with unknown $\pi$ is given by the SEM, and by a permutation matrix $\pi$ (with a slight abuse of notation) applied to $\boldsymbol{X}$. I.e., the flow of information is: $\boldsymbol{N} \xrightarrow{SEM} \boldsymbol{X} \xrightarrow{\pi} \hat{\boldsymbol{X}}$.*

**Lemma A.1** ($\mathbf{J}_{\boldsymbol{f}} \sim_{DAG} (\mathbf{I}_d + \boldsymbol{\mathcal{C}})$). *Given Assum. 1, the partial derivatives of $f_i$ w.r.t. $N_j$ provide information about $\boldsymbol{\mathcal{C}}$, as*

$$(\mathbf{J}_{\boldsymbol{f}})_{kl} = \frac{\partial f_l}{\partial N_k} = 0 \iff \nexists X_k \to \cdots \to X_l$$

*We emphasize that the derivatives are also non-zero in the case of indirect paths, i.e., when $\exists X_i \in p : i \neq k, l$. Furthermore, the strictly lower triangular part of $\mathbf{J}_{\boldsymbol{f}}$ has the describes the same DAG as $\boldsymbol{\mathcal{C}}$–or equivalently, $\mathbf{J}_{\boldsymbol{f}} \sim_{DAG} (\mathbf{I}_d + \boldsymbol{\mathcal{C}})$.*

# B   Identifiability definitions

**Definition B.1** (Strong Identifiability (Khemakhem et al., 2020b)). *Given a parameter class $\Theta$, when the feature extractors $\boldsymbol{g}_{\theta_1}, \boldsymbol{g}_{\theta_2} : \mathcal{X} \to \mathcal{N}$ produce latent representations $\boldsymbol{N}_1 = \boldsymbol{g}_{\theta_1}(\boldsymbol{X}), \boldsymbol{N}_2 = \boldsymbol{g}_{\theta_2}(\boldsymbol{X})$ that are*

*equivalent up to scaled permutations and offsets c for all $\theta_1, \theta_2 \in \Theta$, i.e.,*

$$\theta_1 \sim \theta_2 \iff \boldsymbol{N} = \boldsymbol{g}_{\theta_1}(\boldsymbol{X}) = \mathbf{DP}\boldsymbol{g}_{\theta_2}(\boldsymbol{X}) + c, \tag{12}$$

*where $\mathbf{D}$ is a diagonal and $\mathbf{P}$ a permutation matrix. Then $\theta_1, \theta_2$ fulfill an equivalence relationship.*

**Definition B.2** (Weak Identifiability (Khemakhem et al., 2020b)). *Given a parameter class $\Theta$, when the feature extractors $\boldsymbol{g}_{\theta_1}, \boldsymbol{g}_{\theta_2} : \mathcal{X} \to \mathcal{N}$ produce latent representations $\boldsymbol{N}_1 = \boldsymbol{g}_{\theta_1}(\boldsymbol{X}), \boldsymbol{N}_2 = \boldsymbol{g}_{\theta_2}(\boldsymbol{X})$ that are equivalent up to matrix multiplications and offsets c for all $\theta_1, \theta_2 \in \Theta$, i.e.,*

$$\theta_1 \sim \theta_2 \iff \boldsymbol{N} = \boldsymbol{g}_{\theta_1}(\boldsymbol{X}) = \mathbf{A}\boldsymbol{g}_{\theta_2}(\boldsymbol{X}) + c, \tag{13}$$

*where $\text{rank}(\mathbf{A}) \geq \min(\dim \mathcal{N}; \dim \mathcal{X})$. Then $\theta_1, \theta_2$ fulfill an equivalence relationship.*

**Definition B.3** (Identifiability up to elementwise nonlinearities (Hyvärinen & Morioka, 2017)). *Given a parameter class $\Theta$, when the feature extractors $\boldsymbol{g}_{\theta_1}, \boldsymbol{g}_{\theta_2} : \mathcal{X} \to \mathcal{N}$ produce latent representations $\boldsymbol{N}_1 = \boldsymbol{g}_{\theta_1}(\boldsymbol{X}), \boldsymbol{N}_2 = \boldsymbol{g}_{\theta_2}(\boldsymbol{X})$ that are equivalent up to elementwise nonlinearities, matrix multiplications and offsets c for all $\theta_1, \theta_2 \in \Theta$, i.e.,*

$$\theta_1 \sim \theta_2 \iff \boldsymbol{N} = \boldsymbol{g}_{\theta_1}(\boldsymbol{X}) = \mathbf{A}\sigma\left[\boldsymbol{g}_{\theta_2}(\boldsymbol{X})\right] + c, \tag{14}$$

*where $\text{rank}(\mathbf{A}) \geq \min(\dim \mathcal{N}; \dim \mathcal{X})$ and $\sigma$ denotes an elementwise nonlinear transformation. Then $\theta_1, \theta_2$ fulfill an equivalence relationship.*

## C  Compatibility of SEM–ICA assumptions

Several works investigated the relationship between SEM and ICA (Gresele et al., 2021; Monti et al., 2019; Shimizu et al., 2006; Von Kügelgen et al., 2021; Hyvärinen et al., 2023); however, it is unclear whether and which assumptions of both fields are compatible. This section relies on (Monti et al., 2019, App. B.), where the authors detail the SEM–ICA connection for linear models. The clear difference is that the conventional SEM formulation (Defn. A.1) expresses each $X_i$ as a function of $\boldsymbol{Pa}_i$ and $N_i$; whereas ICA only uses $N_i$. Formally:

$$X_i := f_i(\boldsymbol{Pa}_i, N_i), \qquad \forall i \in \mathcal{I} = \{, \ldots, d\} \tag{15}$$

$$X_i := f_i^*(\boldsymbol{N}^i), \qquad \forall i \in \mathcal{I} = \{0, \ldots, d-1\}, \tag{16}$$

where the former is the conventional definition (Defn. A.1), whereas the latter is a reduced form of the SEM (Defn. A.2, with $\boldsymbol{N}^i$ denoting a subset of $\boldsymbol{N}$, i.e., $\boldsymbol{N}^i \subseteq \boldsymbol{N}$), corresponding to the ICA model. Note that we use an asterisk to denote that the $f_i$ of the two equations *can be different.*

### C.1  Bijectivity of $f$

It is common to assume a *bijective* map from the causes (sources) to the effects (observations) in both the causality (Khemakhem et al., 2021; Gresele et al., 2021; Monti et al., 2019) and the ICA (Zimmermann et al., 2021; Von Kügelgen et al., 2021; Shimizu et al., 2006; Gresele et al., 2021) literatures. However, since the maps in (15) and (16) are not necessarily the same, we need to investigate whether those assumptions are compatible.

**Proposition 3** (Equivalence of bijectivity in SEMs and ICA). *Assuming bijectivity of $f_i(\boldsymbol{Pa}_i, N_i)$ and that of $f_i^*(\boldsymbol{N}^i)$ are equivalent.*

*Proof.* For the proof, we will use an inductive argument and, w.l.o.g., assume that each $X_i$ depends on all $N_{j \leq i}$ (if there are less dependencies, those arguments can be omitted).

$f_i \implies f_i^*$  We start from the conventional SEM equations (Defn. A.1):

$$X_1 := f_1(N_1) \tag{17}$$
$$X_2 := f_2(X_1, N_2) \tag{18}$$

Visually, the question is whether the blue and the red arrows commute (blue are assumed to be bijective, the red's bijectivty needs to be proven):

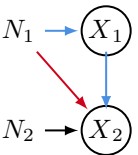

By assumption, $f_1$ is bijective (in $N_1$), and so is $f_2$ (in $X_1$ and $N_2$). Since $f_1 \equiv f_1^*$, we proceed to (18) and substitute (17) into (18), yielding:

$$X_2 := f_2\left(f_1\left(N_1\right), N_2\right). \tag{19}$$

Since the composition of bijective maps is bijective (Macdonald, 1968), so the map from $N_1$ to $X_2$ is bijective, since the maps $N_1 \to X_1$ and $X_1 \to X_2$ are bijective by assumption, yielding the bijectivity of $f_i^*$. Then, we apply the same argument inductively for $X_3 := f_2\left(X_1, X_2, N_3\right)$, and up to $X_d$.

$f_i \Longleftarrow f_i^*$    We start from the reduced SEM equations (Defn. A.2):

$$X_1 := f_1^*\left(N_1\right) \tag{20}$$
$$X_2 := f_2^*\left(N_1, N_2\right). \tag{21}$$

Visually, the question is whether the blue and the red arrows commute (blue are assumed to be bijective, the red's bijectivty needs to be proven):

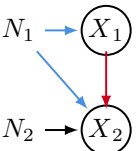

By assumption, $f_1^*$ is bijective (in $N_1$), and so is $f_2^*$ (in $N_1$ and $N_2$). Again, $f_1 \equiv f_1^*$, so we proceed to (21). Since $X_1$ and $N_1$ relate via a bijective map, there is no information lost in the mapping. Thus, using $X_1$ instead of $N_1$ is possible since $f_1^*$ maintains bijectivity—it can be undone by $\left(f_1^*\right)^{-1}$, which exists by assumption. $N_1 \to X_1$ and $N_1 \to X_2$ are bijective maps, decomposing the latter into $N_1 \to X_1 \to X_2$ only implies that $N_1 \to X_1$ is injective and $X_1 \to X_2$ is surjective (Macdonald, 1968). Fortunately, the $N_1 \to X_1$ is bijective by assumption, so we only need to show that $X_1 \to X_2$ is not only surjective, but also bijective. Since both $X_1$ and $X_2$ have the same domain, $X_1 \to X_2$ is bijective (Macdonald, 1968). Then, we apply the same argument inductively for $X_3 := f_2\left(N_1, N_2, N_3\right)$, and up to $X_d$. $\qquad \square$

## C.2   Does identifiability imply no confounders?

Identifiabilty can be thought of as "inverting" the DGP Zimmermann et al. (2021). So the question is whether identifiability implies that the learned representation needs to capture all $N_i$, when the assumptions include that $N_i$ are *jointly independent*? Additionally, we assume that $\dim \boldsymbol{N} = \dim \boldsymbol{X} = d$. Intuitively, if there would be a confounder, it would induce *additional*[6] correlation between at least two $X_p$ and $X_q$. That is, $N_p$ and $N_q$ would need to "emulate" that when $X_k$ changes (via $N_k$), then both $X_p$ and $X_q$ would need to change.

**Proposition 4** (Identifying jointly independent $N_i$ implies no confounders.). *Under the assumption of jointly independent $N_i$ and $\dim \boldsymbol{N} = \dim \boldsymbol{X} = d$, identifiability at least up to elementwise nonlinearities (Defn. B.3) implies that there cannot be confounders.*

*Proof.* We assume that there is a confounder $X_k$, which is the common cause of $X_p$ and $X_q$—the argument generalizes to more children of $X_k$. Two cases emerge: when there is a directed path $X_p \to \cdots \to X_q$ ($p$ and $q$ are interchangeable for our argument), or when there is none.

---

[6]That is, $X_p$ can be the parent of $X_q$, and they still can have another common cause $X_k$

**No directed path between $X_p$ and $X_q$**    Recall from Defn. A.6 that these relationships materialize in the following graph:

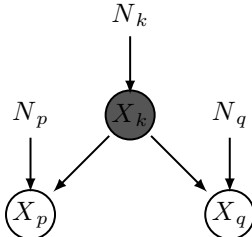

Figure 9: Visualization of a confounder (unobserved common cause), indicated by a gray node color.

From the graph, we can describe the conditional independence relationship of $N_p$ and $N_q$. Namely, we have access to observations $X_p$ and $X_q$, implying ($\perp$ denotes conditional independence):

$$N_p \not\perp N_q | X_p, X_q \quad , \tag{22}$$

since conditioning of $X_p$ and $X_q$ activates the colliders (Defn. A.4) $N_p \to X_p \leftarrow X_k$ and $X_k \to X_q \leftarrow N_q$, the path between $N_p$ and $N_q$ opens up. Thus, $N_p$ and $N_q$ become dependent, contradicting the assumption that $N_p \perp N_q$.

**There is at least one directed path between $X_p$ and $X_q$**    By noticing that conditioning on $X_p$ and $X_q$ blocks any paths between $X_p$ and $X_q$, the conclusion is the same as above.    $\square$

# D  Extended related work

Table 4: Comparison of CD methods. Column $\mathbf{x}$ indicates multivariability, $do\,(\emptyset)$ indicates whether the method can be applied only to observational data, $\boldsymbol{f}$ indicates constraints on the function class of the SEM, $\partial/\partial$ indicates differentiability, and the data column lists restrictions on the data distribution.

| METHOD | $\mathbf{x}$ | $do\,(\emptyset)$ | $\boldsymbol{f}$[7] | $\partial/\partial$ | DATA |
|---|---|---|---|---|---|
| MONTI ET AL. (2019) | ✗ | ✓ | ✓ | ✗ | NON-STATIONARY |
| SHIMIZU ET AL. (2006) | ✓ | ✓ | LINEAR | ✗ | NON-GAUSSIAN |
| GUO ET AL. (2022) | ✓ | ✓ | ✓ | ✗ | EXCHAGEABILITY |
| KHEMAKHEM ET AL. (2021) | ✓ | ✓ | AFFINE/ADDITIVE | ✓ | ✓ |
| LACHAPELLE ET AL. (2020) | ✓ | ✓ | ADDITIVE | ✓ | GAUSSIAN |
| BROUILLARD ET AL. (2020) | ✓ | ✗ | ✓ | ✓ | ✓ |
| KE ET AL. (2020) | ✓ | ✗ | ✓ | ✓ | DISCRETE |
| LIPPE ET AL. (2021) | ✓ | ✗ | ✓ | ✓ | ✓ |
| NG ET AL. (2022) | ✓ | ✓ | ADDITIVE | ✓ | ✓ |
| SCHÖLKOPF ET AL. (2021A) | ✓ | ✓ | LINEAR/ADDITIVE | ✓ | ✓ |
| ZHENG ET AL. (2018) | ✓ | ✓ | LINEAR | ✓ | ✓ |
| YU ET AL. (2019) | ✓ | ✓ | ADDITIVE | ✓ | ✓ |
| SHEN ET AL. (2022)[8] | ✓ | ✓ | ADDITIVE | ✓ | LABELING |
| KALAINATHAN ET AL. (2018) | ✓ | ✓ | ADDITIVE | ✓ | GAUSSIAN |
| ROLLAND ET AL. (2022) | ✓ | ✓ | ADDITIVE | ✓ | ✓ |
| YANG ET AL. (2021)[9] | ✓ | ✓ | ADDITIVE | ✓ | LABELING |
| LORCH ET AL. (2021) | ✓ | ✓[10] | ✓ | ✓ | GRAPH PRIOR |
| LORCH ET AL. (2022) | ✓ | ✓ | ✓ | ✓ | GRAPH PRIOR |
| CHARPENTIER ET AL. (2022) | ✓ | ✓ | ✓ | ✓ | GRAPH PRIOR |
| FARIA ET AL. (2022) | ✓ | ✗[11] | ✓ | GRAPH PRIOR | |
| ZHENG ET AL. (2022) | ✓ | ✓ | LINEAR | ✓ | GAUSSIAN |
| AHUJA ET AL. (2022B) | ✓ | ✓[12] | POLYNOMIAL | ✓ | ✓ |
| SQUIRES ET AL. (2023) | ✓ | ✗ | LINEAR | ✗ | ✓ |
| ATANACKOVIC ET AL. (2023) | ✓ | ✓ | CYCLIC (ODE) | ✓ | ✓ |
| **Ours** | ✓ | ✓ | ✓ | ✓ | ASSUMS. 2 AND F.1 |

Table 5: Using the Jacobian in the literature for CD and/or identifiability. Column $\boldsymbol{f}$ indicates constraints on the function class of the SEM, the data column lists restrictions on the data distribution, $\mathbf{J}$ describes the Jacobian of which function is used, CD indicates use for causal discovery, and the identifiability column whether the method has identifiability guarantees.

| METHOD | $\boldsymbol{f}$ | DATA | $\mathbf{J}$ | CD | IDENTIFIABILITY |
|---|---|---|---|---|---|
| SHIMIZU ET AL. (2006) | LINEAR | NON-GAUSSIAN | $\mathbf{J}_{\boldsymbol{f}^{-1}}$ | ✓ | ✓ |
| LACHAPELLE ET AL. (2020) | ADDITIVE | GAUSSIAN | $\mathbf{J}_{\boldsymbol{f}^{-1}}$ | ✓ | ✗ |
| GRESELE ET AL. (2021)[13] | IMA[14] | ✓ | $\mathbf{J}_{\boldsymbol{f}}$ | ✗ | ✓ |
| ZHENG ET AL. (2022) | SPARSE | ✓ | $\mathbf{J}_{\boldsymbol{f}}$ | ✗ | ✓ |
| ROLLAND ET AL. (2022) | ADDITIVE | ✓ | SCORE FUNCTION | ✓ | ✗ |
| ATANACKOVIC ET AL. (2023) | CYCLIC (ODE) | ✓ | $\mathbf{J}_{\boldsymbol{f}}$ | ✓ | ✗ |
| **Ours** | ✓ | ASSUMS. 2 AND F.1 | $\mathbf{J}_{\boldsymbol{f}^{-1}}$ | ✓ | ✓ |

[7] $\boldsymbol{f}$ is generally assumed to be invertible, but we omitted mentioning it for brevity. That is, ✓ in this column does not necessarily mean no restrictions at all, including our method, which also relies on a bijective $\boldsymbol{f}$

[8] Supervised

[9] Supervised

[10] Lorch et al. (2021) can also leverage interventional data, but it also works from observations

[11] Faria et al. (2022) assume latent intervention targets; known interventions can also be incorporated in a semi-supervised extension

[12] Ahuja et al. (2022b) has stronger identifiability results when interventional data is available

# E  Proofs

## E.1  Proof of Lem. 1

**Lemma 1.** *[DAG DGPs resolve the permutation ambiguity of ICA] When the DGP is a SEM with functional relationships $\boldsymbol{f}$ and an underlying DAG, then the permutation indeterminacy of ICA $\pi_{\mathrm{ICA}}$ can be accounted for such that the Jacobian of the inference network will have a lower-triangular Jacobian, even with unknown causal ordering $\pi$.*

*Proof.* The unknown causal ordering $\pi$ of $N_i$ implies the right-multiplication of $\mathbf{J}_{\boldsymbol{f}^{-1}}$ with $\pi^{-1}$, the permutation indeterminacy of ICA the left-multiplication with $\pi_{\mathrm{ICA}}$, yielding the estimated Jacobian $\mathbf{J}_{\widehat{\boldsymbol{f}}^{-1}}$:

$$\mathbf{J}_{\widehat{\boldsymbol{f}}^{-1}} = \pi_{\mathrm{ICA}} \circ \mathbf{J}_{\boldsymbol{f}^{-1}} \circ \pi^{-1}, \tag{23}$$

where $\pi_{\mathrm{ICA}}$ and $\pi^{-1}$ are not necessarily the same.

If $\pi$ is **unique**, we only need to show that $\pi_{\mathrm{ICA}}$ is also unique. Assume that there exists $\pi_{\mathrm{ICA},1} \neq \pi_{\mathrm{ICA},2}$ such that $\mathbf{J}_{\widehat{\boldsymbol{f}}^{-1}}$ can be transformed into a lower-triangular $\mathbf{J}_{\boldsymbol{f}^{-1}}$ by both. This implies that the rows of $\mathbf{J}_{\widehat{\boldsymbol{f}}^{-1}}$ can be permuted such that it yields a lower-triangular $\mathbf{J}_{\boldsymbol{f}^{-1}}$ (when $\pi$ is already accounted for). Assume that $\pi_{\mathrm{ICA},1}$ yields a lower-triangular $\mathbf{J}_{\boldsymbol{f}^{-1}}$. Then a different $\pi_{\mathrm{ICA},2}$ means that there are at least two rows $i, k$ in $\mathbf{J}_{\widehat{\boldsymbol{f}}^{-1}}$ that can be permuted differently than in $\pi_{\mathrm{ICA},1}$ such that the resulting matrix is still lower-triangular. $\mathbf{J}_{\boldsymbol{f}^{-1}}$ has a non-zero diagonal (cf. the definition of $\mathbf{B}$ in (26)); thus, using a different ordering $\pi_{\mathrm{ICA},2}$ will violate lower-triangularity, for this means that the $i^{th}, k^{th}$ rows after applying $\pi_{\mathrm{ICA},1}$ will be equal to the $k^{th}, i^{th}$ rows of $\pi_{\mathrm{ICA},2}$ (the former being equal to the true Jacobian $\mathbf{J}_{\boldsymbol{f}}$):

$$\left[\pi_{\mathrm{ICA},1}^{-1} \circ \mathbf{J}_{\widehat{\boldsymbol{f}}^{-1}} \circ \pi\right]_{[i,k],:} = \left[\mathbf{J}_{\boldsymbol{f}^{-1}}\right]_{[i,k],:} = \left[\pi_{\mathrm{ICA},2}^{-1} \circ \mathbf{J}_{\widehat{\boldsymbol{f}}^{-1}} \circ \pi\right]_{[k,i],:}, \tag{24}$$

which means that for $\pi_{\mathrm{ICA},2}$ the resulting matrix has nonzero elements at indices $(i,k)$ and $(k,i)$. This violates lower-triangularity, since $k \neq i$, so one of the above means that there is at least one non-zero element above the main diagonal, leading to a contradiction.

If $\pi$ is **not unique**, we can apply the above argument, resulting in a set of permutation matrices, each yielding a lower-triangular Jacobian.  $\square$

## E.2  Proof of Prop. 1

**Proposition 1.** *[$\mathbf{J}_{\boldsymbol{f}^{-1}} \sim_{DAG} (\mathbf{I}_d - \boldsymbol{\mathcal{A}})$]The inverse DGP's Jacobian $\mathbf{J}_{\boldsymbol{f}^{-1}}$ is structurally equivalent to $(\mathbf{I}_d - \boldsymbol{\mathcal{A}})$, when Assum. 1 holds.*

*Proof.* We start from the functional equation of the SEM and note that if $\boldsymbol{X}$ is the input of $\boldsymbol{f}$ (as $\boldsymbol{Pa}_i$ from (1)), then the output is the same $\boldsymbol{X}$ (which deterministically depends on $\boldsymbol{N}$):

$$\boldsymbol{X} = \boldsymbol{X}\left(\boldsymbol{N}\right) := \boldsymbol{f}\left(\boldsymbol{X}\left(\boldsymbol{N}\right), \boldsymbol{N}\right) = \boldsymbol{f}\left(\boldsymbol{X}, \boldsymbol{N}\right). \tag{25}$$

For a given $(\boldsymbol{X}, \boldsymbol{N})$ we can evaluate the Jacobian of $\boldsymbol{f}$ via the chain rule—the key point is that since $\boldsymbol{X}$ is a fix point of $\boldsymbol{f}$, $\mathbf{J}_{\boldsymbol{f}}$ will apprear on both sides (evaluated at the same point, expressed with the bar notation):

$$\mathbf{J}_{\boldsymbol{f}}\big|_{\boldsymbol{X},\boldsymbol{N}} = \frac{\partial \boldsymbol{X}(\boldsymbol{N})}{\partial \boldsymbol{N}}\big|_{\boldsymbol{X},\boldsymbol{N}} = \frac{\partial \boldsymbol{f}(\boldsymbol{X},\boldsymbol{N})}{\partial \boldsymbol{N}}\big|_{\boldsymbol{X},\boldsymbol{N}} = \mathbf{A}\frac{\partial \boldsymbol{X}}{\partial \boldsymbol{N}}\big|_{\boldsymbol{X},\boldsymbol{N}} + \mathbf{B} = \mathbf{A}\mathbf{J}_{\boldsymbol{f}}\big|_{\boldsymbol{X},\boldsymbol{N}} + \mathbf{B} \tag{26}$$

$$\mathbf{A} := \frac{\partial \boldsymbol{f}(\boldsymbol{X},\boldsymbol{N})}{\partial \boldsymbol{X}}\big|_{\boldsymbol{X},\boldsymbol{N}}; \ \mathbf{B} := \frac{\partial \boldsymbol{f}(\boldsymbol{X},\boldsymbol{N})}{\partial \boldsymbol{N}}\big|_{\boldsymbol{X},\boldsymbol{N}}. \tag{27}$$

The above equation can be reordered to yield the expression for $\mathbf{J}_{\boldsymbol{f}}$ (note that $\mathbf{A}$, $\mathbf{B}$ depend on $\boldsymbol{X}$, $\boldsymbol{N}$):

$$\mathbf{J}_{\boldsymbol{f}}\big|_{\boldsymbol{X},\boldsymbol{N}} = (\mathbf{I}_d - \mathbf{A})^{-1}\mathbf{B}, \tag{28}$$

---

[13]Gresele et al. (2021) proposed IMA and showed that it rules out spurious solutions; Buchholz et al. (2022) proved identifiability
[14]That is, $\mathbf{J}_{\boldsymbol{f}}$ has orthogonal columns

where $\mathbf{A}$ describes the $X_i - X_j$ edges in the DAG (i.e., $\mathbf{A} \sim_{DAG} \boldsymbol{\mathcal{A}}$), $\mathbf{B}$ is diagonal (as the $\boldsymbol{X}$ values are fixed). Since we reason about the Jacobian point-wise, we can invoke the inverse function theorem (by assumption, $\boldsymbol{f}$ is bijective) to express $\mathbf{J}_{\boldsymbol{f}^{-1}}$:

$$\mathbf{J}_{\boldsymbol{f}^{-1}} = \mathbf{J}_{\boldsymbol{f}}^{-1} = \mathbf{B}^{-1} \left( \mathbf{I}_d - \mathbf{A} \right). \tag{29}$$

$\mathbf{J}_{\boldsymbol{f}^{-1}} \sim_{DAG} (\mathbf{I}_d - \boldsymbol{\mathcal{A}})$ follows as $\mathbf{A} \sim_{DAG} \boldsymbol{\mathcal{A}}$ and $\mathbf{B}$ is diagonal (the invariance of $\sim_{DAG}$ follows from Def. 1(i)). $\qquad \square$

**Alternative proof**

*Proof.* The proof consists of two steps: 1) leveraging the iterative formulation of the SEM (2), proving that $\mathbf{J}_{\boldsymbol{f}^{-1}} \sim_{DAG} (\mathbf{I}_d - \boldsymbol{\mathcal{A}})$ and 2) relying on the properties of $\sim_{DAG}$ and Lem. 1, showing $\mathbf{J}_{\boldsymbol{f}^{-1}} \sim_{DAG} \mathbf{J}_{\widehat{\boldsymbol{f}}^{-1}}$.

We start by formulating $\mathbf{J}_{\boldsymbol{f}}$ (recall that $\boldsymbol{X} = \boldsymbol{X}^d$) based on the iterative SEM expression (2):

$$\mathbf{J}_{\boldsymbol{f}}\big|_{\boldsymbol{X}^{d-1}, \boldsymbol{N}} = \frac{\partial \boldsymbol{X}^d}{\partial \boldsymbol{N}}\big|_{\boldsymbol{X}^{d-1}, \boldsymbol{N}} = \frac{\partial \boldsymbol{f}(\boldsymbol{X}^{d-1}, \boldsymbol{N})}{\partial \boldsymbol{N}}\big|_{\boldsymbol{X}^{d-1}, \boldsymbol{N}} = \mathbf{A}^{d-1} \frac{\partial \boldsymbol{X}^{d-1}}{\partial \boldsymbol{N}}\big|_{\boldsymbol{X}^{d-1}, \boldsymbol{N}} + \mathbf{B}^{d-1} \tag{30}$$

$$\mathbf{A}^{d-1} := \frac{\partial \boldsymbol{f}(\boldsymbol{X}^{d-1}, \boldsymbol{N})}{\partial \boldsymbol{X}^{d-1}}\big|_{\boldsymbol{X}^{d-1}, \boldsymbol{N}}; \quad \mathbf{B}^{d-1} := \frac{\partial \boldsymbol{f}(\boldsymbol{X}^{d-1}, \boldsymbol{N})}{\partial \boldsymbol{N}}\big|_{\boldsymbol{X}^{d-1}, \boldsymbol{N}}, \tag{31}$$

where $\mathbf{A}$ describes the $X_i - X_j$ edges in the DAG (i.e., $\mathbf{A} \sim_{DAG} \boldsymbol{\mathcal{A}}$), $\mathbf{B}$ is diagonal (as the $\boldsymbol{X}^{d-1}$ values are fixed). Although both $\mathbf{A}, \mathbf{B}$ are dependent from $t$ (superscript), unless $\boldsymbol{f}$ is linear, they are independent when seen through the lens of structural equivalence. By Assum. A.1, it holds that $\mathbf{A}^k \sim_{DAG} \mathbf{A}^j \wedge \mathbf{B}^k \sim_{DAG} \mathbf{B}^j$ : $\forall j, k$. Thus, we will omit superscripts for both.

Realizing that (30) gives us a recursive formula, and recalling that $\boldsymbol{X}^0 = \boldsymbol{0}$ , we can unroll (30) iteratively for $t = d - 1, d - 2, \ldots, 0$:

$$\mathbf{J}_{\boldsymbol{f}} = \mathbf{A} \frac{\partial \boldsymbol{X}^{d-1}}{\partial \boldsymbol{N}} + \mathbf{B} \sim_{DAG} \mathbf{A} \left[ \mathbf{A} \frac{\partial \boldsymbol{X}^{d-2}}{\partial \boldsymbol{N}} + \mathbf{B} \right] + \mathbf{B} \sim_{DAG} \mathbf{A} \left[ \mathbf{A} \left[ \ldots \left[ \mathbf{A} \underbrace{\frac{\partial \boldsymbol{X}^0}{\partial \boldsymbol{N}}}_{=\boldsymbol{0}} + \mathbf{B} \right] \right] + \mathbf{B} \right] + \mathbf{B} \tag{32}$$

$$= \sum_{i=0}^{d-1} \mathbf{A}^i \mathbf{B} = (\mathbf{I}_d - \mathbf{A})^{-1} \mathbf{B}, \tag{33}$$

where the structural equivalences follow by the structural faithfulness of $\mathbf{J}_{\boldsymbol{f}}$ (Assum. A.1), the last equality expresses the sum of the geometric series with elements $\mathbf{A}^i$ (the sum is finite as $\mathbf{A}$ is strictly lower triangular). By invoking the inverse function theorem (by assumption, $\boldsymbol{f}$ is bojective), we can express $\mathbf{J}_{\boldsymbol{f}^{-1}}$:

$$\mathbf{J}_{\boldsymbol{f}^{-1}} = \mathbf{J}_{\boldsymbol{f}}^{-1} \sim_{DAG} \mathbf{B}^{-1} \left( \mathbf{I}_d - \mathbf{A} \right). \tag{34}$$

$\mathbf{J}_{\boldsymbol{f}^{-1}} \sim_{DAG} (\mathbf{I}_d - \boldsymbol{\mathcal{A}})$ follows as $\mathbf{A} \sim_{DAG} \boldsymbol{\mathcal{A}}$ and $\mathbf{B}$ is diagonal (the invariance of $\sim_{DAG}$ follows from Def. 1(i)). $\qquad \square$

# F   NLICA with Contrastive Learning

**Assumption F.1** (Contrastive model on $\mathbb{R}^d$)**.** *We assume that the model satisfies the following conditions:*

   *(i) The encoder is defined as $\widehat{\boldsymbol{f}}^{-1} : \mathcal{X} \to \mathcal{N}'$, where $\mathcal{N}' \subseteq \mathbb{R}^d$ is a convex body (hyperrectangle);*

   *(ii) The conditional distribution $q(\widetilde{\boldsymbol{N}}|\boldsymbol{N})$ associated with our model $\widehat{\boldsymbol{f}}^{-1}$ through $\boldsymbol{h} = \widehat{\boldsymbol{f}}^{-1} \circ \boldsymbol{f}$ is given by $q(\widetilde{\boldsymbol{N}}|\boldsymbol{N}) = C_q^{-1}(\boldsymbol{N}) e^{-\delta \left( \boldsymbol{h}(\widetilde{\boldsymbol{N}}), \boldsymbol{h}(\boldsymbol{N}) \right)/\tau}$ with $C_q(\boldsymbol{N}) := \int e^{-\delta \left( \boldsymbol{h}(\widetilde{\boldsymbol{N}}), \boldsymbol{h}(\boldsymbol{N}) \right)/\tau} d\widetilde{\boldsymbol{N}}$, where $C_q(\boldsymbol{N})$ is the partition function, $\tau > 0$ is a scale parameter, and $\delta$ is the semi-metric from Assum. 2.*

   *(iii) The encoder is trained with a contrastive loss $\mathcal{L}_{\mathrm{CL}}$ using the same $L_{\lceil}\alpha$ metric $\delta$ as in Assum. 2, i.e.,*

$$\mathbb{E}_{\left( \boldsymbol{X}, \widetilde{\boldsymbol{X}}, \boldsymbol{X}^- \right)} \left[ -\log \frac{\exp\left[ -\delta \left( \widehat{\boldsymbol{f}}^{-1}(\boldsymbol{X}), \widehat{\boldsymbol{f}}^{-1}(\widetilde{\boldsymbol{X}}) \right)/\tau \right]}{\exp\left[ -\delta \left( \widehat{\boldsymbol{f}}^{-1}(\boldsymbol{X}), \widehat{\boldsymbol{f}}^{-1}(\widetilde{\boldsymbol{X}}) \right)/\tau \right] + \sum_i^M \exp\left[ -\delta \left( \widehat{\boldsymbol{f}}^{-1}(\boldsymbol{X}^-), \widehat{\boldsymbol{f}}^{-1}(\widetilde{\boldsymbol{X}}) \right)/\tau \right]} \right], \tag{35}$$

   *where $\widetilde{\boldsymbol{X}}$ is the positive pair, $\boldsymbol{X}^-$ are the negative pairs, and $M$ is the number of negative pairs;*

   *(iv) During training one has access to observations $\boldsymbol{X}$, which are samples from these distributions transformed by the generator function (i.e., the SEM) $\boldsymbol{f}$.*

**Are the distributional assumptions for contrastive NLICA testable for CD?** The assumptions on the conditional $p(\widetilde{N}|N)$ and marginal $p(N)$ distributions (Assum. 2) for contrastive NLICA might be deemed peculiar in the context of CL. First, we emphasize that since our results do not require the use of contrastive NLICA, the user is free to chose a different method that guarantees strong identifiability. However, if contrastive NLICA is deemed suitable for a task, then

1. they are neither interfering with assumptions in CD; and
2. they are testable—e.g., by a one-sample Kolmogorov-Smirnov test (Kolmogorov, 1933; Smirnov, 1948).

What we mean by the first point (and elucidate in the next section) that to fulfill Assum. 2, we neither leave nor constrain the function class of $f$.

## G  Experimental details

Table 6: Hyperparameters for our experiments (§ 5)

| PARAMETER | VALUES |
|---|---|
| $\widehat{f}^{-1}$ | 6-LAYER MLP |
| ACTIVATION | LEAKY ReLU |
| BATCH SIZE | 6144 |
| LEARNING RATE | 1e−4 |
| $\mathbb{R}^d$ | $[0;1]^d$ |
| $C_p$ | 1 |
| $m_p$ | 0 |
| $C_{param}$ | 0.05 |
| $m_{param}$ | 1 |
| $p$ | 1 |
| $\tau$ (IN $\mathcal{L}_{\text{CL}}$) | 1 |
| $\alpha$ | 0.5 |

Table 7: Results for **sparse** linear and nonlinear SEMs. Mean Correlation Coefficient (MCC) measures identifiability, $|\mathcal{E}^*|$ is the maximum number of edges in a DAG, Acc is accuracy, Ours is our proposal, HSIC refers to using HSIC independence tests, and SHD is the Structural Hamming Distance

| | | LINEAR | | | | NONLINEAR | | | |
|---|---|---|---|---|---|---|---|---|---|
| $|\mathcal{E}^*|$ | $d$ | MCC | Acc(Ours) | Acc(HSIC) | SHD | MCC | Acc(Ours) | Acc(HSIC) | SHD |
| 6 | 3 | 1. | $0.917_{\pm 0.108}$ | $0.708_{\pm 0.11}$ | 0.111 | 1. | $0.889_{\pm 0.111}$ | $0.75_{\pm 0.144}$ | 0.111 |
| 15 | 5 | $0.961_{\pm 0.062}$ | $0.768_{\pm 0.121}$ | $0.784_{\pm 0.111}$ | $0.256_{\pm 0.132}$ | $0.972_{\pm 0.059}$ | $0.76_{\pm 0.095}$ | $0.84_{\pm 0.098}$ | $0.208_{\pm 0.0873}$ |
| 36 | 8 | $0.844_{\pm 0.184}$ | $0.709_{\pm 0.084}$ | $0.711_{\pm 0.122}$ | $0.322_{\pm 0.109}$ | $0.783_{\pm 0.155}$ | $0.656_{\pm 0.059}$ | $0.708_{\pm 0.119}$ | $0.375_{\pm 0.081}$ |
| 55 | 10 | $0.8_{\pm 0.217}$ | $0.648_{\pm 0.059}$ | $0.715_{\pm 0.1}$ | $0.336_{\pm 0.055}$ | $0.734_{\pm 0.206}$ | $0.618_{\pm 0.044}$ | $0.69_{\pm 0.086}$ | $0.37_{\pm 0.082}$ |

Table 8: Results for **permuted** (i.e., $\pi$ is not the identity) linear and nonlinear SEMs. Mean Correlation Coefficient (MCC) measures identifiability, $|\mathcal{E}^*|$ is the maximum number of edges in a DAG, Acc is accuracy, Ours is our proposal, HSIC refers to using HSIC independence tests, and SHD is the Structural Hamming Distance

| $|\mathcal{E}^*|$ | $d$ | LINEAR | | | | NONLINEAR | | | |
|---|---|---|---|---|---|---|---|---|---|
| | | MCC | Acc(Ours) | Acc(HSIC) | SHD | MCC | Acc(Ours) | Acc(HSIC) | SHD |
| 6 | 3 | 1. | 1. | 0.667 | 0. | 1. | 1. | 0.667 | 0. |
| 15 | 5 | $0.989_{\pm 0.039}$ | $0.949_{\pm 0.098}$ | $0.866_{\pm 0.088}$ | $0.051_{\pm 0.098}$ | $0.988_{\pm 0.039}$ | $0.94_{\pm 0.087}$ | 0.863 | $0.06_{\pm 0.087}$ |
| 36 | 8 | $0.837_{\pm 0.252}$ | $0.834_{\pm 0.162}$ | $0.624_{\pm 0.127}$ | $0.166_{\pm 0.162}$ | $0.752_{\pm 0.232}$ | $0.794_{\pm 0.138}$ | $0.687_{\pm 0.139}$ | $0.206_{\pm 0.138}$ |
| 55 | 10 | $0.852_{\pm 0.251}$ | $0.761_{\pm 0.213}$ | $0.578_{\pm 0.086}$ | $0.239_{\pm 0.213}$ | $0.794_{\pm 0.255}$ | $0.705_{\pm 0.16}$ | $0.573_{\pm 0.05}$ | $0.295_{\pm 0.159}$ |

### G.1 Code for the Sinkhorn operator

```python
import torch
from torch import nn as nn

class SinkhornOperator(object):
    """
    Based on http://arxiv.org/abs/1802.08665
    """

    def __init__(self, num_steps: int):

        if num_steps < 1:
            raise ValueError(f"{num_steps=} should be at least 1")

        self.num_steps = num_steps

    def __call__(self, matrix: torch.Tensor) -> torch.Tensor:
        def _normalize_row(matrix: torch.Tensor) -> torch.Tensor:
            return matrix - torch.logsumexp(matrix, 1, keepdim=True)

        def _normalize_column(matrix: torch.Tensor) -> torch.Tensor:
            return matrix - torch.logsumexp(matrix, 0, keepdim=True)

        S = matrix

        for _ in range(self.num_steps):
            S = _normalize_column(_normalize_row(S))

        return torch.exp(S)
```

Figure 10: PyTorch code for implementing the Sinkhorn operator from (Mena et al., 2018). A Sinklhorn network applies `SinkhornOperator` to the scaled weight matrix $\mathbf{W}/\boldsymbol{\tau}$, where $\boldsymbol{\tau}$ is generally around $1 \cdot 10^{-3}$.

## H   Notation

## Acronyms

**IMA** Independent Mechanism Analysis

**ANM** Additive Noise Model

**CD** Causal Discovery
**CdF** Causal de Finetti
**CL** Contrastive Learning

**DAG** Directed Acyclic Graph
**DGP** Data Generating Process

**i.i.d.** independent and identically distributed
**ICA** Independent Component Analysis
**ICM** Independent Causal Mechanisms

**LiNGAM** Linear Non-Gaussian Acyclic Model **ODE** Ordinary Differential Equation
**LSTM** Long Short-Term Memory

**MCC** Mean Correlation Coefficient **SEM** Structural Equation Model
**MLP** Multi-Layer Perceptron **SHD** Structural Hamming Distance
**SMS** Sparse Mechanism Shift

**NLICA** nonlinear Independent Component Analysis **TCL** Time-Contrastive Learning

## Nomenclature

$\mathcal{L}_\pi$ regularizer for learning $\pi$ $\boldsymbol{Pa}$ parent set of $\boldsymbol{X}$
**S** Sinkhorn network $\boldsymbol{X}$ observation vector
$\mathcal{E}$ edge set of a graph $\mathcal{A}$ adjacency matrix of a SEMs
$\mathcal{L}$ loss function $\mathcal{C}$ connectivity matrix of a SEMs
$\boldsymbol{h}$ composition of encoder and decoder $\boldsymbol{f}$ structural assignment in SEMs
$d$ problem dimensionality $\mathcal{I}$ index set
$\mathcal{N}$ space of the noise variables
$\mathcal{X}$ space of the effect variables
**Algebra** $\pi$ causal ordering
$\alpha$ scalar field $\sim_{DAG}$ structural equivalence
**D** diagonal matrix $f$ a component of $\boldsymbol{f}$
$\mathbf{I}_d$ $d$-dimensional identity matrix **Contrastive Learning**
**J** Jacobian matrix $M$ number of negative samples
**P** permutation matrix $\mathcal{L}_{\text{CL}}$ contrastive loss function
$\widetilde{\boldsymbol{N}}$ positive latent vector
**Causality** $\widetilde{\boldsymbol{X}}$ positive observation vector
$N$ noise (independent) variable component $\boldsymbol{X}^-$ negative observation vector
$X$ observation component $\boldsymbol{\tau}$ temperature in $\mathcal{L}_{\text{CL}}$
$\boldsymbol{N}$ noise (independent) variable vector

