# OpenReview forum: "Jacobian-based Causal Discovery with Nonlinear ICA"
_TMLR — Accepted by TMLR_

### Review · Reviewer_Tqxy · 2023-01-26

**Summary Of Contributions:**

This work considers inferring causal graphs of nonlinear structural causal models of observed variables from non-interventional data under the assumptions including acyclicity and causal sufficiency.

It assumes that the causal graph is identifiable, more specifically, it assumes identifiability conditions provided by existing works including Monti et al. (2019), Shimizu et al. (2006), and Hoyer et al. (2008), Peters et al. (2011).

Then, they propose to estimate the causal graph of such a model using the Jacobian of inverse of the functional assignment f (Inference model). The idea generalizes the linear cases of Shimizu et al. (2006) to nonlinear cases. Shimizu et al. (2006) relates the zero/non-zero pattern of the ICA (Independent Component Analysis) mixing matrix to the existence/non-existence of causal paths. This submission relates the zero/non-zero pattern of the Jacobian to that.

Further, they propose an estimation algorithm generalizing the ICA-based method by Shimizu et al. (2006), i.e., first perform (nonlinear) ICA and second resolve the permutation indeterminacy. They evaluated their claims using simulation experiments.

The relation between the Jacobian and causal structure and the proposed estimation algorithm (especially the algorithm for solving the permutation indeterminacy) would be new knowledge.


**Audience:**

Yes

**Broader Impact Concerns:**

I don’t think this work is related to ethical concerns.

**Claims And Evidence:**

No

**Requested Changes:**

Critical

1. Their model should be more clearly described.

This work says in several places that it assumes the strong identifiability conditions of  Theorem 6 of Zimmermann et al. (2021).  However, Zimmermann et al. (2021) do not directly consider causal models. It should rephrase the conditions using the notations of Eq. (1). I also find some other assumptions in the first ten lines of page 2 and Appendix A. I think those assumptions need to be put together when describing the model considered in Eq. (1).

It is not clear how those assumptions are combined, e.g., assumptions of Zimmermann et al. (2021), bijective functions and faithfulness etc. Possibly, it might happen that some assumptions contradict each other, but it's not very clear since assumptions appear at different places.

Further, the abstract says that it uses non. i.i.d. data, but I could not find where it is described in Eq.(1). The assumptions should be structured in a better way so that readers easily understand what assumptions this proposal needs.

2. Proposition 1: is strongly identified (Khemakhem et al. 2020b. Def.1)

Def. 1 of the paper (Khemakhem et al. 2020b) defines weak identifiability. So, this might be a typo. Further, I would like this submission to have the definition in the paper, not simply citing the paper. Then, this work is more self-contained and it would make it easier for reviewers to see if it is technically correct. Similarly to a previous comment, the condition should be rephrased in their model in Eq. (1). Khemakhem et al. 2020b does not consider causal models directly.

3. This work claims their proposal works for GENERAL nonlinear functions. However, they also need to make assumptions to make the model identifiable. More discussions on to what extent nonlinear functions satisfy all their assumptions are general would be necessary. This comment is also related to the two comments above.

4. It is not clear if their estimation method is consistent. Does the method find the right answer when the sample size goes to infinity? It would be better to clarify whether it is consistent in theory or it is not yet theoretically clear and is implied by simulation experiments.

5. Propositioin 1 in 3.4 (Main result) should have all the assumptions in the statement. Currently, the model assumptions are not clearly stated there.

6. Section 3.5 does not say how the Jacobian is computed or extracted. It should be given explicitly, though it might depend on estimation methods.

7. The definition of the Jacobian J_f^{-1} seems missing, though I could guess.

To strengthen the work

1. Introduction: In contrast, causal models (Pearl, 2009) construct the world according to the Independent Causal Mechanism (ICM) principle.

I’m not sure if Pearl (2009) claims to use the principle. It would be better to specify on which page it says that. Well, maybe, this is a claim by Peters et al. (2017). If so, it would be better to clarify it.

2. Introduction: On the data side, assumptions include non-stationarity (Scholkopf et al., 2012)

I could not find whether Scholkopf et al. (2012) uses non-stationarity for constraining models. It would be better to clarify which page or paragraph it uses non-stationarity as constraint.

3. Lemma 1.: f describes a DAG

I think the meaning of the word “describe” needs to be mathematically explained at some place (Sorry if I missed it!), though I could tell to some extent.

4. Just above Eq. (1): their causal relationship is given by d deterministic functional assignments (Peters et al., 2017)

 Peters et al., 2017 would be better to be replaced by Pearl (2009b). I see that Peters et al. (2017) is a textbook, but it would be better to cite more original books like Pearl (2009b).

5. The definitions of  alpha_d, alpha_u, alpha_l in Eq.(5) seem missing.

6. The definition of L_CL in Algorithm 1 seems missing, though I could tell to some extent.

7. It would be better to explain what is a Sinkhorn network.

8. Caption of Table 1: "paermutation" should be "permutation".

9. Appendix. B: I wonder if Zheng et al., 2018 assumes noise variables are equal as well?

**Strengths And Weaknesses:**

Strengths:

The relation between the Jacobian and causal structure and the proposed estimation algorithm would be new knowledge. Especially the relation of the Jacobian with the causal structure would be interesting to readers since it generalizes many existing major methods, including Shimizu et al. (2006), Hoyer et al. (2008), and Monti et al., 2019, etc.

Overall, the literature review is well-written.

Weaknesses:

Several points on the theoretical results were not very clear, as stated below in the requested changes. This makes it a bit difficult to see the technical correctness.

Some references need to be examined since the reasons for citing them are not very clear or there seems better references.

---

> ### Author Response · Authors · 2023-02-13
> **Response to Reviewer Tqxy**
>
> We thank **Reviewer Tqxy** for assessing our work as *providing new knowledge about the relationship of the Jacobian and causal models* and *a novel algorithm for resolving the permutation indeterminacy*. We are thankful for the *praise of our review of the related work* and for deeming the *investigation of the relationship between nonlinear ICA (particularly through the structure of the inference model's Jacobian) and causal discovery as highly relevant and important.*
> Below, we address the Reviewer's concerns point-by-point.
>
> ## More detailed description of the model and unclear assumptions, including nonlinearity
> We acknowledge that more investigation was needed to assess the compatibility of the assumptions in identifiability (particularly nonlinear ICA) and causal discovery. The reviewer also pointed out that some of the assumptions (e.g., the non-i.i.d. nature of the data) need to be elaborated.
>
> We moved **all formal assumptions into the main text (Ass. 1 for SEMs in Sec. 3.2 and Ass. 2 about the data generating process in Sec. 4.2** and added a discussion about **how contrastive learning induces non-i.i.d. data in the positive pairs and how this enables causal discovery (Sec. 4.1)**  Furthermore, we **assessed the compatibility of nonlinear ICA and causal discovery assumptions in App. C.** To make clear when our Prop. 1 is valid, we included the SEM assumptions in front of the proposition. Since, regarding identifiability, the proof relies only on the notion of identifiability achieved by the algorithm used, we deferred the assumptions for contrastive nonlinear ICA to Sec. 4.
>
> Regarding the function class, since the compatibility assessment does not require further restrictions, the SEM can rely on any nonlinear and bijective map---bijectivity is justified by the assumption that observables and the noise variables have the same dimension and that there cannot be confounders if the noise variables are jointly independent.
>
>
>
> ## Identifiability definitions
> We thank the reviewer for pointing out the reference to the wrong notion of identifiability. We corrected the citation and **added the notions of identifiability to App. B.**
>
>
> ## Consistency of contrastive nonlinear ICA
> As shown by Zimmermann et al. (2021), the **identifiability guarantees hold in the infinite sample limit**.
>
>
> ## Jacobian definition and computation
> We now define $J_{f^{-1}}$ in Sec 3.2 and detail how the inference model's Jacobian is calculated by PyTorch's automatic differentiation module in Sec. 5.1, paragraph ``Inference model".
>
> ## References
> We thank the reviewer for the detailed suggestions on improving our references. We carefully considered each remark and adopted the following changes:
> - For the Independent Causal Mechanisms (ICM) principle, we refer to Peters et al., 2017;
> - We removed the reference to Schölkopf et al., 2012, since it does not rely on non-stationarity;
> - We refer to Pearl (2009b) as the foundational textbook; and
> - We revisited Zheng et al., 2018 and confirmed that the only explicit assumption on the noise distributions is that "We do not assume that $z$ is Gaussian," which we interpret as the method having no distributional restriction.
>
> ## Clarifications, corrections
> We are grateful for the reviewer's detailed feedback, based on which we made the following corrections:
> - We defined and color-coded $\alpha_u, \alpha_c, \alpha_l$ in Eq. (5);
> - We defined the contrastive loss in Eq. (6); and
> - We explained Sinkhorn networks in Sec. 5.1 (paragraph "Inference model") and added PyTorch code for implementing the key operation of the Sinkhorn networks in App. H.1 (Fig. 8).

---

> > ### Comment · Reviewer_Tqxy · 2023-02-18
> > **ICM**
> >
> > What this paper calls ICM for example in Section 4.1 seems the concept of the autonomous mechanism of Pearl (2009). If so, it would be better to use the term autonomous mechanism rather than ICM. ICM sounds more like the criterion used by Daniusis et al. (UAI2010).

---

> > > ### Author Response · Authors · 2023-02-20
> > > **Reference added to autonomous mechanisms**
> > >
> > > We thank the reviewer for pointing out this critical detail. We added a footnote stating that the term autonomous mechanism might be more appropriate.

---

> > ### Comment · Reviewer_Tqxy · 2023-02-18
> > **Assumption 2**
> >
> > I'm interested in how this paper chooses to use the assumptions of Zimmermann et al. (2021) rather than other works (Hyvärinen & Morioka, 2016; 2017; Morioka et al., 2021; Monti et al., 2019). Are those of Zimmermann et al. (2021) milder than the others?
> >
> > Also, this paper says that they can absorb the assumption of uniform distributions into f. Does this mean that f is not identifiable if the noise distributions are unknown?

---

> > > ### Author Response · Authors · 2023-02-20
> > > **Comparing Assumption 2 to other ICA assumptions**
> > >
> > >
> > > ## Comparing Assumption 2 to other ICA assumptions
> > > Since other ICA methods concern different tasks, the assumptions are not directly comparable. We can say that regarding the identifiable algorithms that rely on variable pairs, our assumptions are comparable.
> > > We added a review of the literature regarding methods (including contrastive learning algorithms) that use variable pairs for causal discovery, which is often the case, as the discussion in the recently-published Causal Triplet paper [3] pinpoints, such as in [2,4,5]. Compared to [2], [4], and [5], which only admit changing a subset of variables in the positive pair, our method does not assume that some latent factors remain constant (i.e., they are not augmented); however, in our model, each component has the same conditional distribution. **What we would like to stress that using contrastive learning for identifying causal factors in not unprecedented: [1] show competitive performance on the Causal3DIdent dataset.** We added the discussion to Sec 4.2 (paragraph "Compatibility of CL and CD").
> > >
> > >
> > >
> > > ## Identifiability of $f
> > > The fact that we can absorb the assumption into $f$ means that the causal graph can be extracted even if the marginal differs from a uniform distribution. As we described in "Differences between identifying the DAG and $f$," for extracting the graph, we do not need to know $f$, only structural properties of the Jacobian of $f^{-1}$.
> > >
> > > ## References
> > > [1] Guo, S., Tóth, V., Schölkopf, B., Huszár, F. (2022). Causal de Finetti: On the identification of invariant causal structure in exchangeable data. arXiv preprint arXiv:2203.15756.
> > > [2] Von Kügelgen, J., Sharma, Y., Gresele, L., Brendel, W., Schölkopf, B., Besserve, M., & Locatello, F. (2021). Self-supervised learning with data augmentations provably isolates content from style. Advances in neural information processing systems, 34, 16451-16467.
> > > [3] Liu, Y., Alahi, A., Russell, C., Horn, M., Zietlow, D., Schölkopf, B., & Locatello, F. (2023). Causal Triplet: An Open Challenge for Intervention-centric Causal Representation Learning. arXiv preprint arXiv:2301.05169.
> > > [4] Brehmer, J., De Haan, P., Lippe, P., & Cohen, T. (2022). Weakly supervised causal representation learning. Advances in neural information processing systems.
> > > [5] Locatello, F., Poole, B., Rätsch, G., Schölkopf, B., Bachem, O., & Tschannen, M. (2020, November). Weakly-supervised disentanglement without compromises. In ICML (pp. 6348-6359). PMLR.

---

> ### Public Comment · ~Claude_Ross1 · 2023-12-04
> **Pearl's**
>
> For instance, Pearl's (2009) autonomous mechanism notion appears to be what this study refers to as ICM in Section 4.1. Instead of "ICM," the phrase "autonomous mechanism" would be more appropriate in this case https://moto-x3m.io . Daniusis et al. (UAI2010) utilized ICM as their criteria, which makes more sense.

---

### Review · Reviewer_Skoz · 2023-01-30

**Summary Of Contributions:**

This paper investigates the use of nonlinear ICA for multivariate causal discovery. The authors prove that under strong identifiability, the Jacobian of inference function captures the sparsity structure of the causal graph. This generalizes the classic LiNGAM method for causal discovery to the nonlinear case. Experiment results on synthetic datasets are given which show that the proposed method outperforms previous methods.

**Audience:**

Yes

**Claims And Evidence:**

No

**Requested Changes:**

The detailed comments are provided above. In my opinion, several key points are:
- Transfer the required assumptions for identifiability result of nonlinear ICA used in the paper to the causal discovery setting and provide justification on why those assumptions make sense in causal discovery setting. Also provide intuition on how to interpret them.
- The use of "general nonlinear functions" could be potentially misleading. See detailed comment above.
-  Since various parts such as introduction claimed that scalability is improved, experiments on larger number of variables should be included.

**Strengths And Weaknesses:**

## Strengths:
- The paper studies the connection of nonlinear ICA and causal discovery, which is an interesting and important direction.
-  This paper is well written.
-  The proposed estimation method is reasonable.

## Weaknesses:
-  As mentioned by the authors, the main contributions, i.e., Lem. 1 and Prop. 1, are direct extensions of theorems in LiNGAM. However, it seems that the theoretical difference is not surprising and has been proposed in several previous works. Specifically, Lem. 1 removes the permutation indeterminacy of ICA via the DAG constraint of the causal graph; Prop. 1 is directly based on the standard formulation of structural equation models. Thus, the only difference appears to be the generalization from the linear mixing function (matrix) to the Jacobian of the nonlinear mixing function, which has been widely applied in previous methods as discussed in this manuscript.
- Based on point (1), the key theoretical challenge of building the connection between ICA and causal discovery in the general nonlinear setting lies in the identifiability of nonlinear ICA, especially, in how to make sense of the assumption needed in nonlinear ICA in the setting of causal discovery. In this manuscript, contrastive NLICA is applied for the identification of the mixing function. For it to be identifiable, it is assumed that the ground-truth noises follow a uniform distribution and the conditional distribution is a Laplace (Assum. E.1). Without further justification, the reasonableness of these distributional assumptions may not be convincing in the context of causal discovery.  Even if we apply other methods in nonlinear ICA with identifiability guarantees, the transfer of their assumptions is nontrivial. For example, according to the manuscript, one needs to start from the Jacobian of the inverse of the mixing function to build the connection, where the assumptions on the sources or mixing function are likely to be untestable after the inverse operation or others.

Therefore, I hold the opinion that the current manuscript lacks an important part of the work on the consistency/transferability of the required assumptions in both ICA and causal discovery and the justification for it. Assuming strong identifiability of nonlinear ICA without sufficient attention to the assumptions may oversimplify the problem and make it a rather direct extension of LiNGAM. Since causal discovery is a special case of ICA, the challenge of nonlinear ICA does not lie in the connection between the formulas of them but in the appropriate assumptions for the identifiability of nonlinear ICA, specifically, transferring the assumptions from ICA to those in causal discovery, or proposing testable assumptions on SEMs, which has not been well solved in the current manuscript.

## Other comments:
- The term "general nonlinear functions" in title, abstract, and other sections could be potentially misleading. As Section 2 mentioned, the identifiability result of nonlinear ICA relies on (1) auxiliary variables, or (2) restricting the nonlinear functions (i.e., with no auxiliary variables). As far as I know, results relying on auxiliary variables can have more "general" nonlinear functions (e.g. Hyvärinen et al. (2019)), while results without auxiliary variables tend to have somewhat restrictive conditions on the nonlinear functions (e.g. https://arxiv.org/abs/2208.06406, https://arxiv.org/abs/2206.10044, https://arxiv.org/abs/2206.07751). If the authors intend to focus on auxiliary variables (e.g. domain/environment index), I would suggest emphasizing auxiliary variables (or non-i.i.d. data) in the title and throughout the paper (because this is quite different from standard nonlinear causal discovery setup assuming i.i.d. data). If the authors intend to focus on the case without auxiliary variables, then using "general nonlinear functions" is not accurate. If the authors do not intend to focus on either case, then possibly "Multivariable Causal Discovery with Nonlinear ICA" could be an option, and also not focusing on "general nonlinear functions" throughout the paper.
- What is the definition of accuracy for causal discovery? (SHD seems to be more widely used.)
- Experiments without knowing causal ordering are more realistic in practice. Therefore, SHD should be included in Table 1.
- The experiment section should include more causal discovery baselines that can handle nonlinear functions, e.g., PC, GES, or continuous optimization based methods.
- Since various parts such as introduction claimed that scalability is improved, experiments on larger number of variables should be included. Currently the experiment section only considers 10 variables at most.
- Assumption E.1: What is $\tilde{N}$?
- Abstract mentioned about the assumption of non-i.i.d. data but is not elaborated throughout the paper.
- Footnote of Page 2: I think the authors intended to cite Lorch et al. (2022).
- The method used Sinkhorn networks to account for permutation. This has been proposed by https://arxiv.org/abs/2203.08509 and should be referenced.

---

> ### Author Response · Authors · 2023-02-13
> **Response to Reviewer Skoz**
>
> We thank **Reviewer Skoz** for the detailed and constructive feedback of our work. We appreciate the reviewer's assessment of *the connection of nonlinear ICA and causal discovery as important and interesting*. In the following, we address each aspect of the review.
>
> ## Novelty of the claims
> First, we acknowledge that our representation of the paper's novelty was unclear. We made it explicit in the revision why our work is, to the best of our knowledge, a novel contribution to the field. The reviewer correctly pointed out that our contribution is generalizing previous works (especially LiNGAM[1]) to nonlinear functions by relying on the Jacobian of the inference model. Indeed, Jacobian-based methods are prevalent in the literature; however, as we showcase in the revised version, **our work applies the Jacobian under assumptions and for purposes that in such manner were not considered in the literature.**
>
> Besides the “Using the Jacobian” paragraph in Sec. 6, we compare relevant Jacobian-based methods in Tab. 5. The key takeaway is that **our work is unique in using the inference model's Jacobian without requiring linearity or additivity of the function class (but requiring non-i.i.d. data) to deliver guarantees on causal discovery via identifiability.**
>
> ## Compatibility of assumptions in nonlinear ICA and SEMs, non-i.i.d. data
> We thank the reviewer for pointing out the need for a more detailed investigation of the compatibility of assumptions in nonlinear ICA and causal discovery. We elaborate on this matter in two parts.
> First, we showcase **how the assumptions underlying contrastive nonlinear ICA enable causal discovery by inducing non-i.i.d. data in the positive pairs**. We illustrate this with **Ex. 2 in Sec. 4.1** and point out the connection to the Causal de Finetti theorem of Guo et al. (2022)[2].
>
> Second, we investigate the general **widely-used assumptions of bijectivity (present in both fields) and that of no confounders in App. C**. We show that assuming a bijective generative model for nonlinear ICA is compatible with bijectivity in the SEM and verify that jointly independent latent sources imply no confounders (we assume the same dimensionality for observables and sources).
>
> ## Clarification on general nonlinear functions
> We thank the reviewer for suggesting that the phrasing "general nonlinear functions" can be misleading. Based on the suggestion, we **changed the title to "Jacobian-based Causal Discovery with Nonlinear ICA"** and made it explicit in our contributions that we rely on a form of auxiliary information via the positive pairs, and this constraint makes identifiability possible.
>
> ## Experiments
> We thank the reviewer's suggestions on improving the experimental section of our submission. We added the requested SHD metric to Tab. 1. and defined the accuracy we use to quantify the ratio of correctly identified edges. Additionally, we clarified our claims regarding scalability by stating that our method's practical advantage is not requiring exponentially many conditional independence tests, but not that it scales to hundreds of variables.
>
> We acknowledge that further comparison would strengthen the empirical evaluation of the manuscript. We now note in Sec. 7 ("Limitations" paragraph) that, unfortunately, since the assumptions required for different causal discovery methods are related in a non-obvious way, designing a task where the assumptions are not mutually exclusive, is complicated, especially since contrastive nonlinear ICA assumes the conditional distribution generating the positive pairs. That is, here, the comparison is more complicated than, e.g., when comparing methods that make the same distributional assumptions but rely on different functional constraints (such as NOTEARS from Zheng et al. (2018)[3] and DAG-GNN from Yu et al. (2019)[4]).
>
> ## Corrections
> We are grateful for the reviewer's detailed feedback, based on which we made the following corrections:
> - We corrected the reference in the footnote on page two to Lorch et al. (2022);
> - We defined $\widehat{\mathbf{N}}$ as the noise (source) variables in the augmented sample of the positive pair;
> - We cite Charpentier et al. (2022) when discussing Sinkhorn networks.
>
> ## References
> - [1] Shimizu, S., Hoyer, P. O., Hyvärinen, A., Kerminen, A.,  Jordan, M. (2006). A linear non-Gaussian acyclic model for causal discovery. Journal of Machine Learning Research, 7(10).
> - [2] Guo, S., Tóth, V., Schölkopf, B.,  Huszár, F. (2022). Causal de Finetti: On the identification of invariant causal structure in exchangeable data. arXiv preprint arXiv:2203.15756.
> - [3] Zheng, X., Aragam, B., Ravikumar, P. K.,  Xing, E. P. (2018). Dags with no tears: Continuous optimization for structure learning. Advances in neural information processing systems, 31.
> - [4] Yu, Y., Chen, J., Gao, T.,  Yu, M. (2019, May). DAG-GNN: DAG structure learning with graph neural networks. In International Conference on Machine Learning (pp. 7154-7163). PMLR.

---

> > ### Comment · Reviewer_Skoz · 2023-02-17
> > **Comment**
> >
> > Thanks for the detailed response, which addressed some of my concerns. My further comments are as follows:
> > - I appreciate the example and discussion in Section 4. I still share the same concern as Reviewer qviF that some of the assumptions may  not be well justified/interpreted in the context of causal discovery, especially Assumption 2(v). I also politely disagree with the authors' further response to Reviewer qviF--specifically, although Prop.2 is agnostic to the the nonlinear ICA method, in my opinion it is still important to at least demonstrate that the identifiability assumptions made by some particular nonlinear ICA method can be well justified/interpreted (or do not contradict with each other) in the causal discovery setting.
> > - In the current form, I still find that the auxiliary information assumed by Assumption 2(v) may not be well justified in the causal discovery setting, specifically the way they are related via the conditional distribution. More discussion would be helpful. Also, can $N_i$ and $\tilde{N}_i$ be caused by a separate latent environment module/variable, say $\theta_e$, which is done by Guo et al. (2022)?
> > - As mentioned, although Prop.2 is agnostic to the the nonlinear ICA method, the authors should mention that $f$ should be bijective for "Ours" in Table 5.

---

> > > ### Author Response · Authors · 2023-02-20
> > > **Adding related works using contrastive learning/data pairs for causal discovery**
> > >
> > > ## Justification of assumptions
> > > We thank the reviewer for the constructive feedback. We added a review of the literature regarding methods (including contrastive learning algorithms) that use variable pairs for causal discovery, which is often the case, ass the discussion in the recently-published Causal Triplet paper [5] pinpoints, such as in  [4,6,7]. Compared to [4], [6], and [7], which only admit changing a subset of variables in the positive pair, our method does not assume that some latent factors remain constant (i.e., they are not augmented); however, in our model, each component has the same conditional distribution.
> > > **What we would like to stress, that using contrastive learning for identifying causal factors in not unprecedented: [3] show competitive performance of the Causal3DIdent dataset.** We added the discussion to Sec 4.2 (paragraph "Compatibility of CL and CD").
> > >
> > > We would like to emphasize that methods using nonlinear ICA for causal discovery have demonstrated their practical applicability, e.g., the NonSENS method [2] (which inspired our work) was capable of reconstructing the causal graph from hippocampal fMRI data or in [5], where contrastive learning was deployed to identify the latent factors from augmented sample pairs.
> > > Interestingly, based on the findings of [1] that natural temporal sequences can be described with the family of (generalized) Laplace distributions, there is justification that Assum. 2(v), i.e., prescribing the conditional distribution, is present in real-world tasks.
> > >
> > >
> > > Additionally, we need to acknowledge that contrastive ICA was a feasible choice since it has an open-source implementation. Despite our best efforts, we could not access the implementation details of other nonlinear ICA methods. Besides our proposal being agnostic to the method guaranteeing identifiability and practical connections, the straight-forward access of the codebase all made contrastive ICA the most reasonable choice. Although we agree with the reviewer that a wider range of experiments could add to the contributions of our work, we honestly do not feel that impelementing other ICA algorithms from scratch could fundamentally improve our contribution.
> > >
> > > ## Environment variables and Causal de Finetti
> > > We thank the reviewer for pointing out this relevant case. As we noted in Sec. 4.1, if one follows the original formulation of contrastive learning, the positive pair consists of *two augmented samples* with corresponding latents  $\tilde{N}_i^1, \tilde{N}_i^2$. In our case, following the practical implementation of contrastive methods (i.e., using one augmentation as described by the graph used in our Ex. 2), the original graph of $\tilde{N}_i^1\leftarrow N\rightarrow \tilde{N}_i^2$ is replaced by (given that $\tilde{N}_i^1\leftarrow N$, i.e., the one augmentation is no augmentation) $N\rightarrow \tilde{N}_i$. The situation the reviewer asks for is theoretically more elegant since that case exactly corresponds to the setting of the Causal de Finetti theorem [3].
> > > Since we intended to describe the in practice used variant of contrastive learning, we used a different model (as shown in Ex. 2). In this case, one can think of the environment variable being equal to $N$ (i.e., the same as one sample in the positive pair), which is a special case.
> > >
> > > ## Bijectivity
> > > We thank the reviewer for this important remark, we added a remark to the table, explicitly indicating that our method assumes a bijective $f$ (and also indicate that other, especially ICA-based, methods also assume the same).
> > >
> > > ## Referneces
> > > - [1] Klindt, D. A., Schott, L., Sharma, Y., Ustyuzhaninov, I., Brendel, W., Bethge, M., & Paiton, D. Towards Nonlinear Disentanglement in Natural Data with Temporal Sparse Coding. In ICLR.
> > > - [2] Monti, R. P., Zhang, K., & Hyvärinen, A. (2020, August). Causal discovery with general non-linear relationships using non-linear ica. In Uncertainty in artificial intelligence (pp. 186-195). PMLR.
> > > - [3] Guo, S., Tóth, V., Schölkopf, B.,  Huszár, F. (2022). Causal de Finetti: On the identification of invariant causal structure in exchangeable data. arXiv preprint arXiv:2203.15756.
> > > - [4] Von Kügelgen, J., Sharma, Y., Gresele, L., Brendel, W., Schölkopf, B., Besserve, M., & Locatello, F. (2021). Self-supervised learning with data augmentations provably isolates content from style. Advances in neural information processing systems, 34, 16451-16467.
> > > - [5] Liu, Y., Alahi, A., Russell, C., Horn, M., Zietlow, D., Schölkopf, B., & Locatello, F. (2023). Causal Triplet: An Open Challenge for Intervention-centric Causal Representation Learning. arXiv preprint arXiv:2301.05169.
> > > - [6] Brehmer, J., De Haan, P., Lippe, P., & Cohen, T. (2022). Weakly supervised causal representation learning. Advances in neural information processing systems.
> > > - [7] Locatello, F., Poole, B., Rätsch, G., Schölkopf, B., Bachem, O., & Tschannen, M. (2020, November). Weakly-supervised disentanglement without compromises. In ICML (pp. 6348-6359). PMLR.

---

### Review · Reviewer_qviF · 2023-01-30

**Summary Of Contributions:**

This manuscript proposes a causal discovery method for general nonlinear function, which is based on nonlinear ICA. The theoretical results are based on the assumption that the identifiability of nonlinear ICA has been given. Compared to the previous work, in my perspective, the major contribution lies in the usage of the Jacobian to represent the sparsity structure in the nonlinear case. As described in detail below, it is less clear to me whether the contribution is significant enough. Besides, there exist some fundamental challenges that have not been addressed in the current manuscript.

**Audience:**

Yes

**Broader Impact Concerns:**

No concerns on the ethical implications

**Claims And Evidence:**

Yes

**Requested Changes:**

Points 1 and 2 in the list of weaknesses, especially 2, are my biggest concerns. I would like to recommend it for acceptance if these can be appropriately clarified/addressed.

**Strengths And Weaknesses:**

**Strengths:**

- The writing is especially clear and well-organized.
- The introduction of the background is very detailed and helpful.
- Related works are summarized comprehensively and precisely. Table 4 is so beautiful.
- The considered problem, i.e., multivariable causal discovery with general nonlinear function, is very important both theoretically and practically.

**Weaknesses:**

Although I appreciate the presentation of the results, there exist some nontrivial problems that need to be addressed:

(1) The difference between the proposed theoretical results and those in lingam [1] appears to be not significant.
* The main idea of Lemma 1 in this manuscript is almost the same as Appendix A in the lingam paper [1].  Both use the assumption of DAG structure (i.e., lower-triangular matrix) to resolve the permutation indeterminacy of ICA. The only difference in the proof is that, in this manuscript, the linear mixing function is replaced by the Jacobian of the nonlinear mixing function, which has been widely applied in previous works, such as [2].
 * Proposition 1 in this manuscript is directly based on the formulations of ICA and Causal Discovery (SEMs). Similarly, it seems that the major novelty also comes from the usage of the Jacobian for the nonlinear mixing function.
Although the connection between the proposed method and lingam has been clearly stated in this manuscript, the difference between them does not seem to be significant enough, especially when the major difference (i.e., the usage of Jacobian) has already been proposed before.

(2) It is not clear whether the assumptions for the identifiability of nonlinear ICA are testable in causal discovery. Because SEM is a special case of ICA with additional assumptions like faithfulness and DAG, the connection between the formulations of ICA and SEM is rather straightforward. Thus, the most important challenge is to justify that the assumptions on the ICA side are testable after being transferred to the setting of causal discovery, which has not been addressed in the current manuscript:
* The strong identifiability of the nonlinear ICA has been directly assumed without further discussion on the required assumptions. Actually, those assumptions may not make sense in causal discovery without appropriate justification. For example, in order for the mixing function to be identifiable, assumption E.1 is required and it includes various constraints, such as the marginal distribution of the noises needs to be uniform. These assumptions may not be reasonable in the context of causal discovery. Additional discussions are necessary for the required assumptions.
* As mentioned by the author, the proposed method works with any nonlinear ICA method as long as the identifiability of the ICA model can be guaranteed. However, the connections between ICA and SEM require certain transformations, which makes the justification of the assumptions transferred from one task to another especially difficult. As mentioned in the manuscript, the connection between ICA and causal discovery is based on the structural equivalence between $J_{f^{-1}}$ and $I-A$, where $f$ is the mixing function of ICA and $A$ is the adjacency matrix of the causal graph. Then the assumptions of ICA (e.g., those on the mixing functions) may not make sense after taking these transformations (e.g., the inverse operation).

Therefore, in order to support the theoretical claims of this manuscript, one needs to show that the assumptions are testable in the context of causal discovery. The assumptions for the identifiability of the nonlinear ICA are the most important part of the considered problem and should be emphasized in the manuscript. Assuming the identifiability of nonlinear ICA directly may oversimplify the problem.

---

[1] Shimizu, S., Hoyer, P. O., Hyvärinen, A., Kerminen, A., & Jordan, M. (2006). A linear non-Gaussian acyclic model for causal discovery. Journal of Machine Learning Research, 7(10).

[2] Gresele, L., Von Kügelgen, J., Stimper, V., Schölkopf, B., & Besserve, M. (2021). Independent mechanism analysis, a new concept?. Advances in neural information processing systems, 34, 28233-28248.

---

> ### Author Response · Authors · 2023-02-13
> **Response to Reviewer qviF**
>
> We thank **Reviewer qviF** for the constructive feedback that significantly improved the manuscript. We are thankful for the *praise of our review of the related work* and appreciate that the reviewer deemed our problem setting of *multivariable causal discovery as highly relevant and important.* In the following, we respond to each aspect of the review one by one.
>
> ## Examining novelty, using the inference model's Jacobian
> We acknowledge that our representation of the paper's novelty was unclear. The review pointed out that our Lem. 1. is almost identical to the App. A of the LiNGAM paper---which we explicitly acknowledge as being our motivation. However, an important difference is that our results hold for nonlinear functions.
> Additional criticism comes from the Jacobian-based approach having already been proposed---particularly in Independent Mechanism Analysis (IMA). We addressed these concerns by **extending the discussion on how the Jacobian is used throughout the identifiability and causal discovery literatures (Sec. 6 and App. D)**. There, we show that although relying on first-order approximations is prevalent in the literature, to the best of our knowledge, **our paper is the first, which
> - uses the Jacobian of the inference model
> - for causal models
> - without constraining the function class (but using non-i.i.d. data)
> - with identifiability guarantees.**
>
> Compared to IMA[2], the clear difference is that **IMA restricts the functional relationships** to the IMA class (i.e., to functions with column-orthogonal Jacobian). Furthermore, **IMA[2] does not make claims about causal discovery**--- although it is inspired by the Independent Causal Mechanisms (ICM) principle from the causality literature. Furthermore, IMA[2] leverages the Jacobian of the *generative model*, while our method relies on the Jacobian of the *inference model*.
>
> ## Compatibility of assumptions in nonlinear ICA and SEMs
>  We thank the reviewer for pointing out that the compatibility of the assumptions in nonlinear ICA and causal discovery needs further investigation. We extended the manuscript and also reviewed the changes in the following.
>
>  Since the expression for the generative model in ICA and the SEM has slightly different forms, we investigate whether **assuming bijectivity** of the functional relationships in one case is compatible with the other. We prove compatibility in both directions for bijective maps in **Prop. 3 in App. C.1**.
>
>  We investigate how assumptions on the source (noise) variables (i.e., joint independence in most cases) relate to the **no-confounder** assumption in **Prop. 4. in App. C.2** and show that when the observed and source variables have the same dimensionality, identifiability implies no confounders.
>
>
> ## References
> - [1] Shimizu, S., Hoyer, P. O., Hyvärinen, A., Kerminen, A.,  Jordan, M. (2006). A linear non-Gaussian acyclic model for causal discovery. Journal of Machine Learning Research, 7(10).
> - [2] Gresele, L., Von Kügelgen, J., Stimper, V., Schölkopf, B.,  Besserve, M. (2021). Independent mechanism analysis, a new concept?. Advances in neural information processing systems, 34, 28233-28248.

---

> > ### Comment · Reviewer_qviF · 2023-02-17
> > **Response to the authors**
> >
> > Thanks for the detailed reply and updates. Some of my concerns remain:
> >
> > (1) Regarding the compatibility of assumptions in nonlinear ICA and SEM, the authors discussed the independent noise and bijectivity assumptions. But I respectively disagree that the significant part of the compatibility of identifiability of nonlinear ICA and SEM lies there. In fact, other assumptions for the identifiability of nonlinear ICA are much more necessary to be well justified. In assumption 2, various constraints on the data-generating process are summarized and, all of these, from my perspective, are also constraints on the SEM. In the current manuscript, there still lacks a discussion on these assumptions, e.g., Assumption 2(iv) requires all noises to be of uniform distribution and Assumption 2(v) connects the conditional distribution to rotational invariance and Laplace distribution. Please let me know if I missed anything.
> >
> > (2) I appreciate the added discussion about previous methods applying Jacobian. At the same time, I am still not fully convinced that the claimed novelty is sufficiently significant. But I am not sure whether that concern is important, especially after the change of the title and structure of the manuscript.

---

> > > ### Author Response · Authors · 2023-02-17
> > > **Response to Reviewer qviF (Part II)**
> > >
> > > We thank the reviewer for the quick feedback on the revised manuscript.
> > >
> > > We stress that the distributional assumptions are only relevant if one uses contrastive learning to identify the exogenous variables. However, none of our contributions are tied to this particular nonlinear ICA method. Namely, **Prop.2 is agnostic to the algorithm as long as strong identifiability is satisfied**; thus, any appropriate nonlinear ICA method, or more generally, identifiability method can be used. Which identifiability method is appropriate depends on the data and problem at hand.
> > >
> > > In other words, **we only use contrastive learning and the related distributional assumptions for illustration purposes**. This method is nice to work with because it does not require a separate decoder, is robust to deviations from the distributional assumptions [1], and is highly successful on practical natural image datasets if the positive pairs are generated from random augmentations. In other scenarios like in video data, where positive pairs could be neighbouring frames, other distributional assumptions would be more fitting as shown in [2].
> > >
> > > We acknowledge that without context, the distributional assumptions might seem restrictive. To address this, we added two footnotes to Assum. 2, elaborating that
> > > - if a differentiable inverse cumulative density function (CDF) is present, then by **passing a uniform random variable through thge inverse CDF, we can emulate any real-valued random variable, showing that the assumption on the marginal is not strong**;
> > > - **identifiability for contrastive ICA hold beyond the conditional being a Laplace distribution: a generalized normal distribution**, which is a much larger family, can be selected [1]; however, we acknowledge that this still rules out specific SEMs.
> > >
> > > Additionally, we emphasize that our **Ex.2 shows that the conditional independencies enabling causal discovery rely on constructing a positive pair, without any assumption on the distribution**. To highlight the differences, we **added a paragraph to Sec. 4.2 discussing how identifiability of the function class $f$ and that of the DAG are different**. Since the Jacobian is sufficient to recover the DAG but that does not fully describe $f$, it can happen that causal identification is possible, whereas that of $f$ is not.

---

> > > > ### Author Response · Authors · 2023-02-17
> > > > **Response to Reviewer qviF (Part II), references**
> > > >
> > > > - [1] Zimmermann, R. S., Sharma, Y., Schneider, S., Bethge, M., & Brendel, W. (2021, July). Contrastive learning inverts the data generating process. In International Conference on Machine Learning (pp. 12979-12990). PMLR.
> > > > - [2] Klindt, D. A., Schott, L., Sharma, Y., Ustyuzhaninov, I., Brendel, W., Bethge, M., & Paiton, D. Towards Nonlinear Disentanglement in Natural Data with Temporal Sparse Coding. In International Conference on Learning Representations.

---

### Review · Reviewer_2EU2 · 2023-02-10

**Summary Of Contributions:**

The paper proposes using the non-zero entries in the Jacobian of a nonlinear ICA model to uncover the causal relationships among the input variables, as represented by a directed acyclic graph (DAG). The DAG structure is inferred by training two Sinkhorn networks to generate two doubly stochastic matrices to permute the entries of the Jacobian into a lower triangular matrix. The Jacobian is extracted from a previously trained Nonlinear ICA. The work proves that if the data was indeed generated by a DAG related variables without common unobserved causes, and under the standard Markov and Faithfulness assumptions the graph is recoverable from non-zero entries of the Jacobian if the nonlinear ICA is strongly identifiable. The claimed advantage over the previously proposed two stage procedure of recovering the DAG from nonlinear ICA is improved computational complexity and thus scalability. An empirical evaluation on three different synthertic datasets is provided to support the claims.


**Audience:**

Yes

**Broader Impact Concerns:**

no concerns

**Claims And Evidence:**

No

**Requested Changes:**

## See the weaknesses
## Minor
- Consider citing the Sinkhorn networks paper before the first use of the idea in Algorithm 1. The fact, that you use this model to generate $S_{ICA}$ and $S_{\pi}$ could have been mentioned on page 5, when these matrices are introduced. As it stands, Algorithm 1 is confusing, as the Sinkhorn networks appear out of nowhere - we were just calling $S_{ICA}$ and $S_{\pi}$ doubly stochastic matrices and now they turned into networks.
- In Algorithm 1 $K=|S_{ICA} \dots |$ - should not the second $S$ be $S_{\pi}$?
- You may want to check your references to arxiv. For example, on page 1 you cite Zhang et al., 2015 linking to arxiv, but the paper has been published in a peer-reviewed venue.
- It is unclear what "General Nonlinear Functions" in the title refers to. The causal process over the causal DAG? But there's no support for the generality in the paper in this case. If this refers to the nonlinear ICA model, then why is it relevant in the title? In general, the title is unclear.


**Strengths And Weaknesses:**

## Strengths

### An interesting and clever idea
- The combination of deep learning methods, such as continuous relaxation of the Birkhoff polytope and Nonlinear ICA, with the causal discovery is interesting.
### A variety of employed simulations to test the claims
### A well written and structured paper
- The paper and especially the introductory overview and related work is well written and paints a comprehensive image of the current state of the field.

## Weaknesses
### The choice of the metrics for the accuracy measurement is harder to interpret
- Why not use Precision Recall, or Omission and Comission of edges?
### Unsupported efficiency claims
- After the nonlinear ICA Jacobian is extracted, and possibly thresholded, the authors are left with an NP-complete problem of finding the topological ordering of a DAG adjacency matrix (see https://www.sciencedirect.com/science/article/abs/pii/S0020019016300552, for example). The proposed solution is to solved a relaxed problem via an SGD optimization. It is unclear how that is efficient and whether it is better than other approaches to the problem, such as SAT solvers and related engines.
- The experiments are also not convincing. The largest number of variables is 10, which is not particularly indicative of the claimed scalability.
- Moreover the most informative PR curve is only shown for one of the experiments and it is clear that as the number of variables grows the quality of the proposed estimation methodology is declining rapidly.
### Lack of demonstration of robustness to assumption violations
- What happens if the data generation process is not a DAG? For example, either some amount of autocorrelation is added to the system in a vein similar to the stepwise SEM generation or a longer cycle is introduced.
- With small number of variables it may be feasible to set all weights in the SEM DGP or the MLP, however, as the model becomes larger, the relative weights of the interactions will necessarily become small. The same is true for the NLICA model as it gets larger. It is unclear how this will affect the proposed approach.
### Insufficient efficiency demonstration
- The paper lacks a run-time evaluation of the proposed approach, leaving the reader to assume that it may be slow based on the limited number of nodes in the simulations.
### Missing relevant result
- This is not strictly a weakness, but consider citing this relevant paper, that is also relying on the sparsity of the Jacobian to resolve unidentifiabilities in ICA: https://arxiv.org/abs/2206.07751

---

> ### Author Response · Authors · 2023-02-13
> **Response to Reviewer 2EU2**
>
> We thank **Reviewer 2EU2** for the constructive feedback and the assessment of our work as *a clever idea* with a *comprehensive introductory overview*. We address each of the reviewer's concerns below.
>
> ## Evaluation metrics
> We appreciate the reviewer's suggestion of using different metrics (such as precision/recall or omission/commission) for assessing causal discovery. The reason for our choice was to make our experimental results comparable to the literature since most methods use the Structural Hamming Distance (SHD) [1-4]. To make results easier to interpret, we included the accuracy (i.e., the ratio of correctly identified edges).
>
> ## Efficiency
> We would like to thank the **Reviewers Skoz and 2EU2** for pointing out the unclear formulation of our efficiency claims. We corrected the wording in the revision, stating that we claim to eliminate the need for exponentially many independence tests but not that our method is scalable to hundreds of variables. We also point out that the limiting factor is the degrading performance of contrastive nonlinear ICA, which, we hypothesize, might be connected to the practical issues observed with self-supervised (and, particularly, contrastive) methods [5-7].
>
> Runtime evaluation is an important aspect, as pointed out by the reviewer. Since we rely on a single forward pass to calculate the Jacobian of the inference model, the associated cost is not limiting. Particularly, if one is interested in only the presence or absence of the edges, then The Jacobian calculation needs to be done only once.
>
> ## Robustness
> We acknowledge that our method does not ameliorate the problem caused by possibly weak connections, which is also a weakness of existing methods, such as [3,9]. However, in our multilayer experiments up to 5 layers, the ground-truth unmixing's Jacobian had on average between $2e-3$ (1 layer) and $1e-8$ entries (5 layers), and our method turned out to be competitive (Tab. 3).
>
> When the underlying model cannot be represented as a DAG (e.g., there are bidirectional edges), then that can be represented with a DAG with confounders (unobserved nodes), which violates our assumptions, i.e., our method will not work.
>
>
> ## Clarifications, corrections
> We are grateful for the reviewer's detailed feedback, based on which we made the following corrections:
> - We added the relevant reference of [8] and elaborated how it relates to our contributions both in Sec. 6 and Tab. 4. Importantly, [8] concerns identifiability, but not causal discovery;
> - We corrected the notation of the Sinkhorn networks in Alg. 1 and described how we use them to parametrize doubly-stochastic matrices in Sec 3.5; and
> - We changed our wording from general nonlinear functions to emphasize that we leverage non-i.i.d. data
>
> ## References
> - [1] Tong, A., Atanackovic, L., Hartford, J.,  Bengio, Y. Bayesian Dynamic Causal Discovery. In A causal view on dynamical systems, NeurIPS 2022 workshop.
> - [2] Shen, X., Liu, F., Dong, H., Lian, Q., Chen, Z.,  Zhang, T. (2022). Weakly Supervised Disentangled Generative Causal Representation Learning. Journal of Machine Learning Research, 23, 1-55.
> - [3] Lachapelle, S., Brouillard, P., Deleu, T.,  Lacoste-Julien, S. Gradient-Based Neural DAG Learning. In International Conference on Learning Representations.
> - [4] Brouillard, P., Lachapelle, S., Lacoste, A., Lacoste-Julien, S.,  Drouin, A. (2020). Differentiable causal discovery from interventional data. Advances in Neural Information Processing Systems, 33, 21865-21877.
> - [5] Robinson, J., Sun, L., Yu, K., Batmanghelich, K., Jegelka, S.,  Sra, S. (2021). Can contrastive learning avoid shortcut solutions?. Advances in neural information processing systems, 34, 4974-4986.
> - [6] Wang, Y., Zhang, Q., Wang, Y., Yang, J.,  Lin, Z. Chaos is a Ladder: A New Theoretical Understanding of Contrastive Learning via Augmentation Overlap. In International Conference on Learning Representations.
> - [7] Wen, Z.,  Li, Y. (2021, July). Toward understanding the feature learning process of self-supervised contrastive learning. In International Conference on Machine Learning (pp. 11112-11122). PMLR.
> - [8] Zheng, Y., Ng, I.,  Zhang, K. On the Identifiability of Nonlinear ICA: Sparsity and Beyond. In Advances in Neural Information Processing Systems.
> - [9] Shimizu, S., Hoyer, P. O., Hyvärinen, A., Kerminen, A.,  Jordan, M. (2006). A linear non-Gaussian acyclic model for causal discovery. Journal of Machine Learning Research, 7(10).

---

### Author Response · Authors · 2023-02-13
**Response to all Reviewers 1/2**

We thank all reviewers for their constructive feedback that significantly improved the manuscript. We appreciate the reviewers' assessment of our work as *providing new knowledge about the relationship of the Jacobian and causal models* (**Reviewer Tqxy**), *a novel algorithm for resolving the permutation indeterminacy* (**Reviewer Tqxy**), and *a clever idea* (**Reviewer 2EU2**). We are thankful for the *praise of our review of the related work* by **Reviewers Tqxy, qviF, and 2EU2**. We appreciate that **Reviewers Tqxy, qviF, and Skoz** deemed our problem setting of *investigating the relationship between nonlinear ICA (particularly through the structure of the inference model's Jacobian) and causal discovery as highly relevant and important.*


## Compatibility of assumptions in nonlinear ICA and SEMs

 All reviewers pointed out that the compatibility of the assumptions in nonlinear ICA and causal discovery needs to be further investigated. In the revised manuscript, we review the main assumptions and show their compatibility.

 Since the expression for the generative model in ICA and the SEM has slightly different forms, we investigate whether **assuming bijectivity** of the functional relationships in one case is compatible with the other. We prove compatibility in both directions for bijective maps in **Prop. 3 in App. C.1**.

 We investigate how assumptions on the source (noise) variables (i.e., joint independence in most cases) relate to the **no-confounder** assumption in **Prop. 4. in App. C.2** and show that in the case when the observed and source variables have the same dimensionality, identifiability implies no confounders.

We also include a more detailed description of the assumptions for contrastive nonlinear ICA in Sec. 4.2, with further discussion about testability and the description of our inference model in **App. G.** Additionally, we illustrate how contrastive learning enables causal discovery in Sec. 4.1, which connects our work to the Causal de Finetti theorem [6].


## Improving self-containedness
To ease the reader's journey, we include additional definitions for causal structures in **App. A** and define the **notions of identifiability in App. B.**

## Examining novelty
**Reviewer Tqxy** stated that our work **relating the Jacobian and the causal structure would be new knowledge** and also deemed our approach for **resolving the permutation indeterminacy novel**.
However, **Reviewers qviF and Skoz** questioned the novelty of some of our claims. We acknowledge that our representation of the paper's novelty was unclear. Besides addressing each reviewer's concerns one by one, here we summarize the new insights of our work.

**Reviewer Skoz** stated that our main results, **Lem. 1, Props. 1 and 2** rely on the conventional formulation of Structural Equation Models (SEMs). This was our intent, i.e., to frame our proposal compatible with previous works. **Reviewer qviF** pointed out that our Lem. 1. is almost identical to the App. A of the LiNGAM [1] paper---which we explicitly acknowledge as our motivation. However, an important difference is that our results hold for nonlinear functions (both LiNGAM and our method have distributional assumptions). Additional criticism comes from the Jacobian-based approach having already been proposed---particularly in Independent Mechanism Analysis (IMA) [2]. IMA, however, uses the Jacobian for identifiability of the ground-truth latents and is not concerned with causal discovery. Hence, IMA has the same goal as NLICA. We address these concerns by **extending the discussion on how the Jacobian is used throughout the identifiability and causal discovery literatures (Sec. 6 and App. D)**. There, we show that although relying on first-order approximations is prevalent in the literature, to the best of our knowledge, **our paper is the first**, which
- uses the Jacobian of the inference model
- for causal models
- without constraining the function class (but using non-i.i.d. data)
- with identifiability guarantees.

Compared to IMA[2], the clear difference is that **IMA restricts the functional relationships** to the IMA class (i.e., to functions with column-orthogonal Jacobian). Furthermore, **IMA[2] does not make claims about causal discovery**--- although it is inspired by the Independent Causal Mechanisms (ICM) principle from the causality literature. Furthermore, IMA[2] leverages the Jacobian of the *generative model*, while our method relies on the Jacobian of the *inference model*.

---

> ### Author Response · Authors · 2023-02-13
> **Response to all Reviewers 2/2**
>
> ## Efficiency
> We would like to thank the **Reviewers Skoz and 2EU2** for pointing out the unclear formulation of our efficiency claims. We corrected the wording in the revision, stating that we claim to eliminate the need for exponentially many independence tests but not that our method is scalable to hundreds of variables. We also point out that the limiting factor is the degrading performance of contrastive nonlinear ICA, which, we hypothesize, might be connected to the practical issues observed with self-supervised (and, particularly, contrastive) methods [3-5].
>
> ## References
> - [1] Shimizu, S., Hoyer, P. O., Hyvärinen, A., Kerminen, A.,  Jordan, M. (2006). A linear non-Gaussian acyclic model for causal discovery. Journal of Machine Learning Research, 7(10).
> - [2] Gresele, L., Von Kügelgen, J., Stimper, V., Schölkopf, B.,  Besserve, M. (2021). Independent mechanism analysis, a new concept?. Advances in neural information processing systems, 34, 28233-28248.
> - [3] Robinson, J., Sun, L., Yu, K., Batmanghelich, K., Jegelka, S.,  Sra, S. (2021). Can contrastive learning avoid shortcut solutions?. Advances in neural information processing systems, 34, 4974-4986.
> - [4] Wang, Y., Zhang, Q., Wang, Y., Yang, J.,  Lin, Z. (2022) Chaos is a Ladder: A New Theoretical Understanding of Contrastive Learning via Augmentation Overlap. In International Conference on Learning Representations.
> - [5] Wen, Z.,  Li, Y. (2021). Toward understanding the feature learning process of self-supervised contrastive learning. In International Conference on Machine Learning (pp. 11112-11122). PMLR.
> - [6] Guo, S., Tóth, V., Schölkopf, B.,  Huszár, F. (2022). Causal de Finetti: On the identification of invariant causal structure in exchangeable data. arXiv preprint arXiv:2203.15756.

---

### Decision · Action_Editors · 2023-03-15

**Recommendation:** Accept with minor revision

**Comment:**

This is a nicely written work that develops a novel use of the Jacobian of the inference network for causal discovery.

The reviewers have concerns about the impact of this work, as the current paper (with emphasis on contrastive NLICA, at least in the paper's current form) struggles with the above-mentioned identifiability assumptions for contrastive NLICA. Also, the authors do not focus on other forms of NLICA, so despite their mentioning that this work need not be restricted to contrastive NLICA, the audience is not left with an alternative. The impact (and also significance) would be greater depending on how identifiability results for contrastive NLICA can be extended in the future; while expected impact is not a criterion for acceptance, significance of results certainly is.

Despite concerns regarding assumptions (which also affects significance), I believe the positives carry enough weight where this work is worthy of publication. The use of the Jacobian of the inference model in particular is novel. I am willing to give the authors the benefit of the doubt on the point of the identifiability assumptions and hope that there will be more discussion about this point going forward. I strongly suggest that the authors read the reviews again and so as to better respond to the criticisms of Reviewers Skoz and qviF. To quote one reviewer’s comment to me:
>  “it is important to demonstrate that the identifiability assumptions made by the particular nonlinear ICA method (chosen by the authors) can be well justified/interpreted in the causal discovery setting”.

And paraphrasing a second point from another comment:
> While the authors assume independence and invertibility, it is well-known that nonlinear ICA is not identifiable without additional assumptions. Hence, more discussion of additional assumptions should be included, and these assumptions should be motivated in the causal discovery setting.

This is an acceptance with a minor revision. I request the authors revise their discussion to better address the above two points; one possibility is to focus on a nonlinear ICA method other than contrastive nonlinear ICA (I am not asking you to do experiments with such a method) in this discussion, if that makes things easier for you, and given that minor revision, I welcome the acceptance of this paper.

**Audience:**

All reviewers agree that what this work sets out to do is of interest, and, modulo identifiability discussion above and below, there is a significant audience for this work.

**Claims And Evidence:**

The main persisting criticism after the authors’ revisions is regarding the identifiability assumptions, i.e., identifiability of nonlinear ICA (NLICA), with the emphasis being on the justification of identifiability of contrastive NLICA in the setting of causal discovery. I believe the other criticisms of the work were sufficiently well-addressed in the revisions and have no other concerns regarding claims and evidence. However, regarding claims and evidence, both appear to be present now that the authors have significantly revised their work. I leave the evaluation of their significance (this is where the importance of assumptions comes in) to the "Comment" section below.

---

> ### Author Response · Authors · 2023-04-07
> **Response to the Decision**
>
> Dear Action Editor,
>
> We appreciate the favorable decision and would like to thank you for your efforts in managing our submission.
>
> In the camera-ready version, we addressed the remaining concerns to the best of our knowledge, particularly:
> - We added references to the *Compatibility of CL and CD* paragraph in 4.2 to showcase how contrastive learning can be deployed for causal discovery, in which we cite the recent work of [1] that connects the contrastive paradigm (particularly, having access to positive pairs) and information about mechanism shifts, which is a well-known hypothesis in the causality literature
> - We added a new paragraph *CD with identifiability beyond ICA* to the Discussion (Sec 7) discussing TCL and sparse mechanism shifts [2], highlighting how other nonlinear ICA methods can fit into the framework of causality
>
> Our code and the logs used for creating the figures are also included.
>
>
> ### References
> - [1] Liu, Y., Alahi, A., Russell, C., Horn, M., Zietlow, D., Schölkopf, B., & Locatello, F. (2023). Causal Triplet: An Open Challenge for Intervention-centric Causal Representation Learning. arXiv preprint arXiv:2301.05169.
> - [2] Perry, R., Von Kügelgen, J., & Schölkopf, B. Causal Discovery in Heterogeneous Environments Under the Sparse Mechanism Shift Hypothesis. In Advances in Neural Information Processing Systems.